# DIFFERENTIALLY PRIVATE LOW-DIMENSIONAL SYNTHETIC DATA FROM HIGH-DIMENSIONAL DATASETS

## ABSTRACT

Differentially private synthetic data provide a powerful mechanism to enable data analysis while protecting sensitive information about individuals. However, when the data lie in a high-dimensional space, the accuracy of the synthetic data suffers from the curse of dimensionality. In this paper, we propose a differentially private algorithm to generate low-dimensional synthetic data efficiently from a high-dimensional dataset with a utility guarantee with respect to the Wasserstein distance. A key step of our algorithm is a private principal component analysis (PCA) procedure with a near-optimal accuracy bound that circumvents the curse of dimensionality. Unlike the standard perturbation analysis, our analysis of private PCA works without assuming the spectral gap for the covariance matrix.

## 1 INTRODUCTION

As data sharing is increasingly locking horns with data privacy concerns, privacy-preserving data analysis is becoming a challenging task with far-reaching impact. Differential privacy (DP) has emerged as the gold standard for implementing privacy in various applications (Dwork & Roth, 2014). For instance, DP has been adopted by several technology companies (Dwork et al., 2019) and has also been used in connection with the release of Census 2020 data (Abowd et al., 2022). The motivation behind the concept of differential privacy is the desire to protect an individual's data while publishing aggregate information about the database, as formalized in the following definition:

**Definition 1.1** (Differential Privacy (Dwork & Roth, 2014))**.** *A randomized algorithm $\mathcal{M}$ is $\varepsilon$-differentially private if for any neighboring datasets $D$ and $D'$ and any measurable subset $S \subseteq$* range($\mathcal{M}$), *we have*

$$\mathbb{P}\left\{\mathcal{M}(D) \in S\right\} \leq e^{\varepsilon}\,\mathbb{P}\left\{\mathcal{M}(D') \in S\right\},$$

*where the probability is with respect to the randomness of $\mathcal{M}$.*

However, utility guarantees for DP are usually provided only for a fixed, predefined set of queries. Hence, it has been frequently recommended that differential privacy may be combined with synthetic data to achieve more flexibility in private data sharing (Hardt et al., 2012; Bellovin et al., 2019). Synthetic datasets are generated from existing datasets and maintain the statistical properties of the original dataset. Ideally, synthetic data contain no protected information; hence, the datasets can be shared freely among investigators in academia or industry, without security and privacy concerns.

Yet, computationally efficient construction of accurate differentially private synthetic data is challenging. Most research on private synthetic data has been concerned with counting queries, range queries, or $k$-dimensional marginals, see e.g. (Hardt et al., 2012; Ullman & Vadhan, 2011; Blum et al., 2013; Vietri et al., 2022; Dwork et al., 2015; Thaler et al., 2012; Boedihardjo et al., 2022c). Notable exceptions are (Wang et al., 2016; Boedihardjo et al., 2022b; Donhauser et al., 2023). Specifically, (Boedihardjo et al., 2022b) provides utility guarantees with respect to the 1-Wasserstein distance. Invoking the Kantorovich-Rubinstein duality theorem, the 1-Wasserstein distance accuracy bound ensures that all Lipschitz statistics are preserved uniformly. Given that numerous machine learning algorithms are Lipschitz (von Luxburg & Bousquet, 2004; Kovalev, 2022; Bubeck & Sellke, 2021; Meunier et al., 2022), this provides data analysts with a vastly increased toolbox of machine learning methods for which one can expect similar outcomes for the original and synthetic data.

For instance, for the special case of datasets living on the $d$-dimensional Boolean hypercube $[0, 1]^d$ equipped with the Hamming distance, the results in (Boedihardjo et al., 2022b) show that there exists an $\varepsilon$-DP algorithm with an expected utility loss that scales like

$$\left(\log(\varepsilon n)^{\frac{3}{2}} / (\varepsilon n)\right)^{1/d}, \tag{1.1}$$

where $n$ is the size of the dataset. While (He et al., 2023) succeeded in removing the logarithmic factor in (1.1), it can be shown that the rate in (1.1) is otherwise tight. Consequently, the utility guarantees in (Boedihardjo et al., 2022b; He et al., 2023) are only useful when $d$, the dimension of the data, is small (or if $n$ is exponentially larger than $d$). In other words, we are facing the curse of dimensionality. The curse of dimensionality extends beyond challenges associated with Wasserstein distance utility guarantees. Even with a weaker accuracy requirement, the hardness result from Uhlman and Vadhan (Ullman & Vadhan, 2011) shows that $n = \text{poly}(d)$ is necessary for generating DP-synthetic data in polynomial time while maintaining approximate covariance.

In (Donhauser et al., 2023), the authors succeeded in constructing DP synthetic data with utility bounds where $d$ in (1.1) is replaced by $(d' + 1)$, assuming that the dataset lies in a certain $d'$-dimensional subspace. However, the optimization step in their algorithm exhibits exponential time complexity in $d$, see (Donhauser et al., 2023, Section 4.1).

This paper presents a computationally efficient algorithm that does not rely on any assumptions about the true data. We demonstrate that our approach enhances the utility bound from $d$ to $d'$ in (1.1) when the dataset is in a $d'$-dimensional affine subspace. Specifically, we derive a DP algorithm to generate low-dimensional synthetic data from a high-dimensional dataset with a utility guarantee with respect to the 1-Wasserstein distance that captures the intrinsic dimension of the data.

Our approach revolves around a private principal component analysis (PCA) procedure with a near-optimal accuracy bound that circumvents the curse of dimensionality. Different from classical perturbation analysis (Chaudhuri et al., 2013; Dwork et al., 2014) that utilizes the Davis-Kahan theorem (Davis & Kahan, 1970) in the literature, our accuracy analysis of private PCA works without assuming the spectral gap for the covariance matrix.

**Notation** In this paper, we work with data in the Euclidean space $\mathbb{R}^d$. For convenience, the data matrix $\mathbf{X} = [X_1, \ldots, X_n] \in \mathbb{R}^{d \times n}$ also indicates the dataset $(X_1, \ldots, X_n)$. We use $\mathbf{A}$ to denote a matrix and $v, X$ as vectors. $\| \cdot \|_F$ is the Frobenius norm and $\| \cdot \|$ is the operator norm of a matrix, respectively. Two sequences $a_n, b_n$ satisfies $a_n \lesssim b_n$ if $a_n \leq C b_n$ for an absolute constant $C > 0$.

**Organization of the paper** The rest of the paper is arranged as follows. In the remainder of Section 1, we present our algorithm with an informal theorem for privacy and accuracy guarantees in Section 1.1, followed by a discussion. A comparison to the state of the art is given in Section 1.2. Next, we consider the Algorithm 1 step by step. Section 2 discusses private PCA and noisy projection. In Section 3, we modify synthetic data algorithms from (He et al., 2023) to the specific cases on the lower dimensional spaces. The precise privacy and accuracy guarantee of Algorithm 1 is summarized in Section 4. We provide additional useful lemmas and definitions in Section A. Section B contains more details about the low-dimensional synthetic data step in Algorithm 1. Proofs are contained in Section C. Finally, since the case $d' = 1$ is not covered in Theorem 1.2, we discuss additional results under stronger assumptions in Section D.

## 1.1 MAIN RESULTS

In this paper, we use Definition 1.1 on data matrix $\mathbf{X} \in \mathbb{R}^{d \times n}$. We say two data matrices $\mathbf{X}, \mathbf{X}'$ are *neighboring datasets* if $\mathbf{X}$ and $\mathbf{X}'$ differ on only one column. We follow the setting and notation in (He et al., 2023) as follows. let $(\Omega, \rho)$ be a metric space. Consider a dataset $\mathbf{X} = [X_1, \ldots, X_n] \in \Omega^n$. We aim to construct a computationally efficient differentially private randomized algorithm that outputs synthetic data $\mathbf{Y} = [Y_1, \ldots, Y_n] \in \Omega^m$ such that the two empirical measures

$$\mu_{\mathbf{X}} = \frac{1}{n} \sum_{i=1}^{n} \delta_{X_i} \quad \text{and} \quad \mu_{\mathbf{Y}} = \frac{1}{m} \sum_{i=1}^{m} \delta_{Y_i}$$

are close to each other. Here $\delta_{X_i}$ denotes the Dirac measure centered on $X_i$. We measure the utility of the output by $\mathbb{E} W_1(\mu_{\mathbf{X}}, \mu_{\mathbf{Y}})$, where the expectation is taken over the randomness of the algorithm.

We assume that each vector in the original dataset $\mathbf{X}$ is inside $[0,1]^d$; our goal is to generate a differentially private synthetic dataset $\mathbf{Y}$ in $[0,1]^d$, where each vector is close to a linear subspace of dimension $d'$, and the empirical measure of $\mathbf{Y}$ is close to $\mathbf{X}$ under the 1-Wasserstein distance. We introduce Algorithm 1 as a computationally efficient algorithm for this task. It can be summarized in the following four steps:

1. Construct a private covariance matrix $\widehat{\mathbf{M}}$. The private covariance is constructed by adding a Laplacian random matrix to a centered covariance matrix $\mathbf{M}$ defined as

$$\mathbf{M} = \frac{1}{n-1}\sum_{i=1}^{n}(X_i - \overline{X})(X_i - \overline{X})^{\mathsf{T}}, \quad \text{where} \quad \overline{X} = \frac{1}{n}\sum_{i=1}^{n}X_i. \tag{1.2}$$

   This step is presented in Algorithm 2.

2. Find a $d'$-dimensional subspace $\widehat{\mathbf{V}}_{d'}$ by taking the top $d'$ eigenvectors of $\widehat{\mathbf{M}}$. Then, project the data onto a linear subspace. The new data obtained in this way are inside a $d'$-dimensional ball. This step is summarized in Algorithm 3.

3. Generate a private measure in the $d'$ dimensional ball centered at the origin by adapting methods in (He et al., 2023), where synthetic data generation algorithms were analyzed for data in the hypercube. This is summarized in Algorithms 4 and 5.

4. Add a private mean vector to shift the dataset back to a private affine subspace. Given the transformations in earlier steps, some synthetic data points might lie outside the hypercube. We then metrically project them back to the domain of the hypercube. Finally, we output the resulting dataset $\mathbf{Y}$. This is summarized in the last two parts of Algorithm 1.

The next informal theorem states the privacy and accuracy guarantees of Algorithm 1. Section 4 gives more detailed and precise statements.

**Theorem 1.2.** *Let $\Omega = [0,1]^d$ equipped with $\ell^\infty$ metric and $\mathbf{X} = [X_1,\ldots,X_n] \in \Omega^n$ be a dataset. For any $2 \le d' \le d$, Algorithm 1 outputs an $\varepsilon$-differentially private synthetic dataset $\mathbf{Y} = [Y_1,\ldots,Y_m] \in \Omega^m$ for some $m \ge 1$ in polynomial time such that*

$$\mathbb{E}W_1(\mu_{\mathbf{X}},\mu_{\mathbf{Y}}) \lesssim_d \sqrt{\sum_{i>d'}\sigma_i(\mathbf{M})} + (\varepsilon n)^{-1/d'}, \tag{1.3}$$

*where $\lesssim_d$ means the right hand side of (1.3) hides factors that are polynomial in $d$, and $\sigma_i(\mathbf{M})$ is the $i$-th eigenvalue value of $\mathbf{M}$ in (1.2).*

Note that $m$, the size of the synthetic dataset $\mathbf{Y}$, is not necessarily equal to $n$ since the low-dimensional synthetic data subroutine in Algorithm 1 creates noisy counts. See Section 3 for more details.

**Optimality** The accuracy rate in (1.3) is optimal up to a $\text{poly}(d)$ factor when $\mathbf{X}$ lies in an affine $d'$-dimensional subspace. The second term matches the lower bound in (Boedihardjo et al., 2022b, Corollary 9.3) for generating $d'$-dimensional synthetic data in $[0,1]^{d'}$. The first term is the error from the best rank-$d'$ approximation of $\mathbf{M}$. It remains an open question if the first term is necessary for methods that are not PCA-based. A more detailed discussion can be found below Theorem 4.2.

**Improved accuracy if $X$ is low-dimensional** When the original dataset $\mathbf{X}$ lies in an affine $d'$-dimensional subspace, it implies $\sigma_i(\mathbf{M}) = 0$ for $i > d'$ and $\mathbb{E}W_1(\mu_{\mathbf{X}},\mu_{\mathbf{Y}}) \lesssim_d (\varepsilon n)^{-1/d'}$. This is an improvement from the accuracy rate $O((\varepsilon n)^{-1/d})$ for unstructured data in $[0,1]^d$ in (Boedihardjo et al., 2022b; He et al., 2023), which overcomes the curse of high dimensionality.

**$Y$ is a low-dimensional representation of $X$** The synthetic dataset $\mathbf{Y}$ is close to a $d'$-dimensional subspace under the 1-Wasserstein distance, as shown in Proposition 3.2.

**Adaptive and private choices of $d'$** One can choose the value of $d'$ adaptively and privately based on singular values of $\widehat{\mathbf{M}}$ in Algorithm 2 such that $\sigma_{d'+1}(\widehat{\mathbf{M}})$ is relatively small compared to $\sigma_{d'}(\widehat{\mathbf{M}})$. A near-optimal $d'$ is chosen by balancing the two error terms to find the best trade-off in (1.3). More detailed discussion on its privacy and accuracy can be found in Appendix E

---

**Algorithm 1** Low-dimensional Synthetic Data

---

**Input:** True data matrix $\mathbf{X} = [X_1, \ldots, X_n]$, $X_i \in [0,1]^d$, privacy parameter $\varepsilon$.

**Private covariance matrix** Apply Algorithm 2 to $\mathbf{X}$ with privacy parameter $\varepsilon/3$ to obtain a private covariance matrix $\widehat{\mathbf{M}}$.

**Private linear projection** Choose a target dimension $d'$. Apply Algorithm 3 with privacy parameter $\varepsilon/3$ to project $\mathbf{X}$ onto a private $d'$-dimensional linear subspace. Save the private mean $\overline{X}_{\mathrm{priv}}$.

**Low-dimensional synthetic data** Use subroutine in Section 3 to generate $\varepsilon/3$-DP synthetic data $\mathbf{X}'$ of size $m$ depending on $d' = 2$ or $d' \geq 3$.

**Adding the private mean vector** Shift the data back by $X_i'' = X_i + \overline{X}_{\mathrm{priv}}$.

**Metric projection** Define $f : \mathbb{R} \to [0,1]$ such that

$$f(x) = \begin{cases} 0 & \text{if } x < 0; \\ x & \text{if } x \in [0,1]; \\ 1 & \text{if } x > 1. \end{cases}$$

Then, for $v \in \mathbb{R}^d$, we define $f(v)$ to be the result of applying $f$ to each coordinate of $v$.

**Output:** Synthetic data $\mathbf{Y} = [f(X_1''), \ldots, f(X_m'')]$.

---

**Running time** The *private linear projection* step in Algorithm 1 has a running time $O(d^2 n)$ using the truncated SVD (Li et al., 2019). The *low-dimensional synthetic data* subroutine has a running time polynomial in $n$ for $d' \geq 3$ and linear in $n$ when $d' = 2$ (He et al., 2023). Therefore, the overall running time for Algorithm 1 is linear in $n$, polynomial in $d$ when $d' = 2$ and is $\mathrm{poly}(n, d)$ when $d' \geq 3$. Although sub-optimal in the dependence on $d'$ for accuracy bounds, one can also run Algorithm 1 in linear time by choosing PMM (Algorithm 4) in the subroutine for all $d' \geq 2$.

### 1.2 COMPARISON TO PREVIOUS RESULTS

**Private synthetic data** Most existing work considered generating DP-synthetic datasets while minimizing the utility loss for specific queries, including counting queries Blum et al. (2013); Hardt et al. (2012); Dwork et al. (2009), $k$-way marginal queries Ullman & Vadhan (2011); Dwork et al. (2015), histogram release Abowd et al. (2019). For a finite collection of predefined linear queries $Q$, Hardt et al. (2012) provided an algorithm with running time linear in $|Q|$ and utility loss grows logarithmically in $|Q|$. The sample complexity can be reduced if the queries are sparse (Dwork et al., 2015; Blum et al., 2013; Donhauser et al., 2023).

Beyond finite collections of queries, Wang et al. (2016) considered utility bound for differentiable queries, and recent works (Boedihardjo et al., 2022b; He et al., 2023) studied Lipschitz queries with utility bound in Wasserstein distance. Donhauser et al. (2023) considered sparse Lipschitz queries with an improved accuracy rate. Balog et al. (2018); Harder et al. (2021); Kreacic et al. (2023); Yang et al. (2023) measure the utility of DP synthetic data by the maximum mean discrepancy (MMD) between empirical distributions of the original and synthetic datasets. This metric is different from our chosen utility bound in Wasserstein distance. Crucially, MMD does not provide any guarantees for Lipschitz downstream tasks.

Our work provides an improved accuracy rate for low-dimensional synthetic data generation. Compared to (Donhauser et al., 2023), our algorithm is computationally efficient and has a better accuracy rate. Besides (Donhauser et al., 2023), we are unaware of any work on low-dimensional synthetic data generation from high-dimensional datasets. Our experiments in Section 5 also show the importance of exploring the low-dimensional structure for private synthetic data generation.

While methods from Boedihardjo et al. (2022b); He et al. (2023) can be directly applied if the low-dimensional subspace is known, the subspace would be non-private and could reveal sensitive information about the original data. The crux of our paper is that we do not assume the low-dimensional subspace is known, and our DP synthetic data algorithm protects its privacy.

**Private PCA**    Private PCA is a commonly used technique for differentially private dimension reduction of the original dataset. This is achieved by introducing noise to the covariance matrix (Mangoubi & Vishnoi, 2022; Chaudhuri et al., 2013; Imtiaz & Sarwate, 2016; Dwork et al., 2014; Jiang et al., 2016; 2013; Zhou et al., 2009). Instead of independent noise, the method of exponential mechanism is also extensively explored (Kapralov & Talwar, 2013; Chaudhuri et al., 2013; Jiang et al., 2016). Another approach, known as streaming PCA (Oja, 1982; Jain et al., 2016), can also be performed privately (Hardt & Price, 2014; Liu et al., 2022a).

The private PCA typically yields a private $d'$-dimensional subspace $\widehat{\mathbf{V}}_{d'}$ that approximates the top $d'$-dimensional subspace $\mathbf{V}_{d'}$ produced by the standard PCA. The accuracy of private PCA is usually measured by the distance between $\widehat{\mathbf{V}}_{d'}$ and $\mathbf{V}_{d'}$ (Dwork et al., 2014; Hardt & Roth, 2013; Mangoubi & Vishnoi, 2022; Liu et al., 2022a; Singhal & Steinke, 2021). To prove a utility guarantee, a common tool is the Davis-Kahan Theorem (Bhatia, 2013; Yu et al., 2015), which assumes that the covariance matrix has a spectral gap (Chaudhuri et al., 2013; Dwork et al., 2014; Hardt & Price, 2014; Jiang et al., 2016; Liu et al., 2022a). Alternatively, using the projection error to evaluate accuracy is independent of the spectral gap (Kapralov & Talwar, 2013; Liu et al., 2022b; Arora et al., 2018). In our implementation of private PCA, we don't treat $\widehat{\mathbf{V}}_{d'}$ as our terminal output. Instead, we project $\mathbf{X}$ onto $\widehat{\mathbf{V}}_{d'}$. Our approach directly bound the Wasserstein distance between the projected dataset and $\mathbf{X}$. This method circumvents the subspace perturbation analysis, resulting in an accuracy bound independent of the spectral gap, as outlined in Lemma 2.2.

Singhal & Steinke (2021) considered a related task that takes a true dataset close to a low-dimensional linear subspace and outputs a private linear subspace. To the best of our knowledge, none of the previous work on private PCA considered low-dimensional DP synthetic data generation.

**Centered covariance matrix**    A common choice of the covariance matrix for PCA is $\frac{1}{n}\mathbf{X}\mathbf{X}^{\mathsf{T}}$ (Chaudhuri et al., 2011; Dwork et al., 2014; Singhal & Steinke, 2021), which is different from the centered one defined in (1.2). The rank of $\mathbf{X}$ is the dimension of the linear subspace that the data lie in rather than that of the affine subspace. If $\mathbf{X}$ lies in a $d'$-dimensional affine space (not necessarily passing through the origin), centering the data shifts the affine hyperplane spanned $\mathbf{X}$ to pass through the origin. Consequently, the centered covariance matrix will have rank $d'$, whereas the rank of $\mathbf{X}$ is $d' + 1$. By reducing the dimension of the linear subspace by 1, the centering step enhances the accuracy rate from $(\varepsilon n)^{-1/(d'+1)}$ to $(\varepsilon n)^{-1/d'}$. Yet, this process introduces the added challenge of protecting the privacy of mean vectors, as detailed in the third step in Algorithm 1 and Algorithm 3.

**Private covariance estimation**    Private covariance estimation (Dong et al., 2022; Mangoubi & Vishnoi, 2022) is closely linked to the private covariance matrix and the private linear projection components of our Algorithm 1. Instead of adding i.i.d. noise, (Kapralov & Talwar, 2013; Amin et al., 2019) improved the dependence on $d$ in the estimation error by sampling top eigenvectors with the exponential mechanism. However, it requires $d'$ as an input parameter (in our approach, it can be chosen privately) and a lower bound on $\sigma_{d'}(\mathbf{M})$. The dependence on $d$ is a critical aspect in private mean estimation (Kamath et al., 2019; Liu et al., 2021), and it is an open question to determine the optimal dependence on $d$ for low-dimensional synthetic data generation.

## 2    PRIVATE LINEAR PROJECTION

### 2.1    PRIVATE CENTERED COVARIANCE MATRIX

We start with the first step: finding a $d'$ dimensional private linear affine subspace and projecting $\mathbf{X}$ onto it. Consider the $d \times n$ data matrix $\mathbf{X} = [X_1, \ldots, X_n]$, where $X_1, \ldots, X_n \in \mathbb{R}^d$. The rank of the covariance matrix $\frac{1}{n}\mathbf{X}\mathbf{X}^{\mathsf{T}}$ measures the dimension of the *linear subspace* spanned by $X_1, \ldots, X_n$. If we subtract the mean vector and consider the centered covariance matrix $\mathbf{M}$ in (1.2), then the rank of $\mathbf{M}$ indicates the dimension of the *affine linear subspace* that $\mathbf{X}$ lives in.

To guarantee the privacy of $\mathbf{M}$, we add a symmetric Laplacian random matrix $\mathbf{A}$ to $\mathbf{M}$ to create a private Hermitian matrix $\widehat{\mathbf{M}}$ from Algorithm 2. The variance of entries in $\mathbf{A}$ is chosen such that the following privacy guarantee holds:

**Theorem 2.1.** *Algorithm 2 is $\varepsilon$-differentially private.*

---

**Algorithm 2** Private Covariance Matrix

---

**Input:** Matrix $\mathbf{X} = [X_1, \ldots, X_n]$, privacy parameter $\varepsilon$, and variance parameter $\sigma = \frac{3d^2}{\varepsilon n}$.

**Computing the covariance matrix** Compute the mean $\overline{X} = \frac{1}{n} \sum_{i=1}^n X_i$ and the centered covariance matrix $\mathbf{M}$.

**Generating a Laplacian random matrix** Generate i.i.d. independent random variables $\lambda_{ij} \sim \mathrm{Lap}(\sigma), i \leq j$. Define a symmetric matrix $\mathbf{A}$ such that

$$\mathbf{A}_{ij} = \mathbf{A}_{ji} = \begin{cases} \lambda_{ij} & \text{if } i < j; \\ 2\lambda_{ii} & \text{if } i = j, \end{cases}$$

**Output:** The noisy covariance matrix $\widehat{\mathbf{M}} = \mathbf{M} + \mathbf{A}$.

---

## 2.2 NOISY PROJECTION

The private covariance matrix $\widehat{\mathbf{M}}$ induces private subspaces spanned by eigenvectors of $\widehat{\mathbf{M}}$. We then perform a truncated SVD on $\widehat{\mathbf{M}}$ to find a private $d'$-dimensional subspace $\widehat{\mathbf{V}}_{d'}$ and project original data onto $\widehat{\mathbf{V}}_{d'}$. Here, the matrix $\widehat{\mathbf{V}}_{d'}$ also indicates the subspace generated by its orthonormal columns. The full steps are summarized in Algorithm 3.

---

**Algorithm 3** Noisy Projection

---

**Input:** True data matrix $\mathbf{X} = [X_1, \ldots, X_n]$, $X_i \in [0, 1]^d$, privacy parameters $\varepsilon$, the private covariance matrix $\widehat{\mathbf{M}}$ from Algorithm 2, and a target dimension $d'$.

**Singular value decomposition** Compute the top $d'$ orthonormal eigenvectors $\widehat{v}_1, \ldots, \widehat{v}_{d'}$ of $\widehat{\mathbf{M}}$ and denote $\widehat{\mathbf{V}}_{d'} = [\widehat{v}_1, \ldots, \widehat{v}_{d'}]$.

**Private centering** Compute $\overline{X} = \frac{1}{n} \sum_{i=1}^n X_i$. Let $\lambda \in \mathbb{R}^d$ be a random vector with i.i.d. components of $\mathrm{Lap}(d/(\varepsilon n))$. Shift each $X_i$ to $X_i - (\overline{X} + \lambda)$ for $i \in [n]$.

**Projection** Project $\{X_i - (\overline{X} + \lambda)\}_{i=1}^n$ onto the linear subspace spanned by $\widehat{v}_1, \ldots, \widehat{v}_{d'}$. The projected data $\widehat{X}_i$ is given by $\widehat{X}_i = \sum_{j=1}^{d'} \left\langle X_i - (\overline{X} + \lambda), \widehat{v}_j \right\rangle \widehat{v}_j$.

**Output:** The data matrix after projection $\widehat{\mathbf{X}} = [\widehat{X}_1 \ldots \widehat{X}_n]$.

---

Algorithm 3 only guarantees private basis $\widehat{v}_1, \ldots, \widehat{v}_{d'}$ for each $\widehat{X}_i$, but the coordinates of $\widehat{X}_i$ in terms of $\widehat{v}_1, \ldots, \widehat{v}_{d'}$ are *not private*. Algorithms 4 and 5 in the next stage will output synthetic data on the private subspace $\widehat{\mathbf{V}}_{d'}$ based on $\widehat{\mathbf{X}}$. The privacy analysis combines the two stages based on Lemma A.2, and we state the results in Section 3.

## 2.3 ACCURACY GUARANTEE FOR NOISY PROJECTION

The data matrix $\widehat{\mathbf{X}}$ corresponds to an empirical measure $\mu_{\widehat{\mathbf{X}}}$ supported on the subspace $\widehat{\mathbf{V}}_d$. In this subsection, we characterize the 1-Wasserstein distance between the empirical measure $\mu_{\widehat{\mathbf{X}}}$ and the empirical measure of the centered dataset $\mathbf{X} - \overline{X}\mathbf{1}^\mathsf{T}$, where $\mathbf{1} \in \mathbb{R}^n$ is the all-1 vector. This problem can be formulated as the stability of a low-rank projection based on a covariance matrix with additive noise. We first provide the following useful deterministic lemma.

**Lemma 2.2** (Stability of noisy projection). *Let $\mathbf{X}$ be a $d \times n$ matrix and $\mathbf{A}$ be a $d \times d$ Hermitian matrix. Let $\mathbf{M} = \frac{1}{n}\mathbf{X}\mathbf{X}^\mathsf{T}$ with eigenvalues $\sigma_1 \geq \sigma_2 \geq \cdots \geq \sigma_d$. Let $\widehat{\mathbf{M}} = \frac{1}{n}\mathbf{X}\mathbf{X}^\mathsf{T} + \mathbf{A}$, $\widehat{\mathbf{V}}_{d'}$ be a $d \times d'$ matrix whose columns are the first $d'$ orthonormal eigenvectors of $\widehat{\mathbf{M}}$, and $\mathbf{Y} = \widehat{\mathbf{V}}_{d'}\widehat{\mathbf{V}}_{d'}^\mathsf{T}\mathbf{X}$. Let $\mu_\mathbf{X}$ and $\mu_\mathbf{Y}$ be the empirical measures of column vectors of $\mathbf{X}$ and $\mathbf{Y}$, respectively. Then*

$$W_2^2(\mu_\mathbf{X}, \mu_\mathbf{Y}) \leq \frac{1}{n}\|\mathbf{X} - \mathbf{Y}\|_F^2 \leq \sum_{i>d'} \sigma_i + 2d'\|\mathbf{A}\|. \tag{2.1}$$

Inequality (2.1) holds without any spectral gap assumption on $\mathbf{M}$. In the context of sample covariance matrices for random datasets, a related bound without a spectral gap condition is derived in (Reiss & Wahl, 2020, Proposition 2.2). Furthermore, Lemma 2.2 bears a conceptual resemblance to (Achlioptas & McSherry, 2001, Theorem 5), which deals with low-rank matrix approximation under perturbation. With Lemma 2.2, we derive the following Wasserstein distance bounds between the centered dataset $\mathbf{X} - \overline{X}\mathbf{1}^{\mathsf{T}}$ and the dataset $\widehat{\mathbf{X}}$.

**Theorem 2.3.** *For input data $\mathbf{X}$ and output data $\widehat{\mathbf{X}}$ in Algorithm 3, let $\mathbf{M}$ be the covariance matrix defined in* (1.2). *Then for an absolute constant $C > 0$,*

$$\mathbb{E}W_1(\mu_{\mathbf{X}-\overline{X}\mathbf{1}^{\mathsf{T}}}, \mu_{\widehat{\mathbf{X}}}) \leq \left(\mathbb{E}W_2^2(\mu_{\mathbf{X}-\overline{X}\mathbf{1}^{\mathsf{T}}}, \mu_{\widehat{\mathbf{X}}})\right)^{1/2} \leq \sqrt{2\sum_{i>d'}\sigma_i(\mathbf{M})} + \sqrt{\frac{Cd'd^{2.5}}{\varepsilon n}}.$$

## 3 SYNTHETIC DATA SUBROUTINES

In the next stage of Algorithm 1, we construct synthetic data on the private subspace $\widehat{\mathbf{V}}_{d'}$. Since the original data $X_i$ is in $[0,1]^d$, after Algorithm 3, we have

$$\left\|\widehat{X}_i\right\|_2 = \left\|X_i - \overline{X} - \lambda\right\|_2 \leq \sqrt{d} + \left\|\overline{X} + \lambda\right\|_2 =: R$$

for any fixed $\lambda \in \mathbb{R}^d$. Therefore, the data after projection would lie in a $d'$-dimensional ball embedded in $\mathbb{R}^d$ with radius $R$, and the domain for the subroutine is

$$\Omega' = \{a_1\widehat{v}_1 + \cdots + a_{d'}\widehat{v}_{d'} \mid a_1^2 + \cdots + a_{d'}^2 \leq R^2\},$$

where $\widehat{v}_1, \ldots, \widehat{v}_{d'}$ are the first $d'$ private principal components in Algorithm 3. Depending on whether $d' = 2$ or $d' \geq 3$, we apply two different algorithms from (He et al., 2023). Since the adaptations are similar, the case for $d' \geq 3$ is deferred to Appendix B.

### 3.1 $d' = 2$: PRIVATE MEASURE MECHANISM (PMM)

Algorithm 4 is adapted from the Private Measure Mechanism (PMM) in (He et al., 2023, Algorithm 4). PMM starts with a binary hierarchical partition of a compact domain $\Omega$ of $r$ levels, and it adds inhomogeneous with variance $\sigma_j$ noise to the number of data points in the $j$-th level of all subregions. It then ensures the counts in all regions are nonnegative and the counts of two subregions at level $j$ add up to the count of a bigger region at level $j-1$. Finally, it releases synthetic data according to the noisy counts in each subregion at level $r$. More details about PMM can be found in Appendix B.1.

Since we need a suitable binary partition for the high-dimensional ball $\Omega'$, to reduce to the case studied in (He et al., 2023), we enlarge $\Omega'$ to a hypercube $[-R, R]^{d'}$ inside the linear subspace $\widehat{\mathbf{V}}_{d'}$. The privacy and accuracy guarantees are proved in the next proposition.

---

**Algorithm 4** PMM Subroutine

---

**Input:** dataset $\widehat{\mathbf{X}} = (\widehat{X}_1, \ldots, \widehat{X}_n)$ in the region

$$\Omega' = \{a_1\widehat{v}_1 + \cdots + a_{d'}\widehat{v}_{d'} \mid a_1^2 + \cdots + a_{d'}^2 \leq R\}.$$

**Binary partition** Let $r = \lceil\log_2(\varepsilon n)\rceil$ and $\sigma_j = \varepsilon^{-1} \cdot 2^{\frac{1}{2}(1-\frac{1}{d'})(r-j)}$. Enlarge the region $\Omega'$ into

$$\Omega_{\text{PMM}} = \{a_1\widehat{v}_1 + \cdots + a_{d'}\widehat{v}_{d'} \mid a_i \in [-R, R], \forall i \in [d']\}.$$

Build a binary partition $\{\Omega_\theta\}_{\theta\in\{0,1\}^{\leq r}}$ on $\Omega_{\text{PMM}}$.

**Noisy count** For any $\theta$, count the number of data in the region $\Omega_\theta$ denoted by $n_\theta = \left|\widehat{\mathbf{X}} \cap \Omega_\theta\right|$, and let $n'_\theta = (n_\theta + \lambda_\theta)_+$, where $\lambda_\theta$ are independent integer Laplacian random variables with $\lambda \sim \text{Lap}_{\mathbb{Z}}(\sigma_{|\theta|})$, and $|\theta|$ is the length of the vector $\theta$.

**Consistency** Enforce consistency of $\{n'_\theta\}_{\theta\in\{0,1\}^{\leq r}}$

**Output:** Synthetic data $\mathbf{X}'$ randomly sampled from $\{\Omega_\theta\}_{\theta\in\{0,1\}^r}$.

---

**Proposition 3.1.** *The subroutine Algorithm 4 is $\varepsilon$-differentially private. For any $d' \geq 2$, with the input as the projected data $\widehat{\mathbf{X}}$ and the range $\Omega'$ with radius $R$, Algorithm 4 has an accuracy bound*

$$\mathbb{E}W_1(\mu_{\widehat{\mathbf{X}}}, \mu_{\mathbf{X}'}) \leq CR(\varepsilon n)^{-1/d'},$$

*where the expectation is taken with respect to the randomness of the synthetic data subroutine, conditioned on $R$.*

### 3.2 Adding a private mean vector and metric projection

Since we shift the data by its private mean before projection, we need to add another private mean vector back, which shifts the dataset $\widehat{\mathbf{X}}$ to a new private affine subspace close to the original dataset $\mathbf{X}$. The output data vectors in $\mathbf{X}''$ (defined in Algorithm 1) are not necessarily inside $[0, 1]^d$. The subsequent metric projection enforces all synthetic data inside $[0, 1]^d$. Importantly, this post-processing step does not have privacy costs.

After metric projection, dataset $\mathbf{Y}$ from the output of Algorithm 1 is close to an affine subspace, as shown in the next proposition. Notably, (3.1) shows that the metric projection step does not cause the largest accuracy loss among all subroutines.

**Proposition 3.2** ($\mathbf{Y}$ **is close to an affine subspace**)**.** *The function $f : \mathbb{R}^d \to [0, 1]^d$ is the metric projection to $[0, 1]^d$ with respect to $\| \cdot \|_\infty$, and the accuracy error for the metric projection step in Algorithm 1 is dominated by the error of the previous steps:*

$$W_1(\mu_{\mathbf{Y}}, \mu_{\mathbf{X}''}) \leq W_1(\mu_{\mathbf{X}}, \mu_{\mathbf{X}''}), \tag{3.1}$$

*where the dataset $\mathbf{X}''$ defined in Algorithm 1 is in a $d'$-dimensional affine subspace. And we have*

$$\mathbb{E}W_1(\mu_{\mathbf{Y}}, \mu_{\mathbf{X}''}) \lesssim_d \sqrt{\sum_{i>d'} \sigma_i(\mathbf{M})} + (\varepsilon n)^{-1/d'}.$$

## 4 Privacy and accuracy of Algorithm 1

In this section, we summarize the privacy and accuracy guarantees of Algorithm 1. The privacy guarantee is proved by analyzing three parts of our algorithms: private mean, private linear subspace, and private data on an affine subspace.

**Theorem 4.1** (Privacy)**.** *Algorithm 1 is $\varepsilon$-differentially private.*

The next theorem combines errors from linear projection, synthetic data subroutine using PMM or PSMM, and the post-processing error from mean shift and metric projection.

**Theorem 4.2** (Accuracy)**.** *For any given $2 \leq d' \leq d$ and $n > 1/\varepsilon$, the output data $\mathbf{Y}$ from Algorithm 1 with the input data $\mathbf{X}$ satisfies*

$$\mathbb{E}W_1(\mu_{\mathbf{X}}, \mu_{\mathbf{Y}}) \lesssim \sqrt{\sum_{i>d'} \sigma_i(\mathbf{M})} + \sqrt{\frac{d'd^{2.5}}{\varepsilon n}} + \sqrt{\frac{d}{d'}}(\varepsilon n)^{-1/d'}, \tag{4.1}$$

*where $\mathbf{M}$ denotes the covariance matrix in* (1.2).

There are three terms on the right-hand side of (4.1). The first term is the error from the rank-$d'$ approximation of the covariance matrix $\mathbf{M}$. The second term is the accuracy loss for private PCA after the perturbation from a random Laplacian matrix. The optimality of this error term remains an open question. The third term is the accuracy loss when generating synthetic data in a $d'$-dimensional subspace. Notably, the factor $\sqrt{d/d'}$ is both requisite and optimal. This can be seen by the fact that a $d'$-dimensional section of the cube can be $\sqrt{d/d'}$ times larger than the low-dimensional cube $[0, 1]^{d'}$ (e.g., if it is positioned diagonally). Complementarily, (Boedihardjo et al., 2022b) showed the optimality of the factor $(\varepsilon n)^{-1/d'}$ for generating $d'$-dimensional synthetic data in $[0, 1]^{d'}$. Therefore, the third term in (4.1) is necessary and optimal.

## 5 SIMULATION

In this section, we showcase the empirical results obtained from our Algorithm 1, which produces DP synthetic data based on the Optical Recognition of Handwritten Digits (Alpaydin & Kaynak, 1998). This dataset consists of 5620 images of digits with $8 \times 8$ pixels, represented as vectors in $[0, 1]^{64}$. We split the dataset into 3823 training data and 1797 testing data. The top one in Figure 1 is a random sample of the images in the training set.

Since the labels of the hand-written digits are $\{0, \ldots, 9\}$, we split the database into ten classes according to their labels and apply Algorithm 1 separately with privacy parameter $\varepsilon$. The synthetic images generated in this way have the correct labels automatically. The bottom one in Figure 1 are synthetic images generated by Algorithm 1 with $d' = 4$ and $\varepsilon = 4$. We then combine the synthetic digit images from 10 classes as the *synthetic* training set for the SVM algorithm. It is worth mentioning that the algorithm still gives $\varepsilon$-differential privacy because each image is used only once.

To evaluate the utility of the synthetic dataset, in Figure 2, we apply the trained SVM classifier to the test dataset from Alpaydin & Kaynak (1998) and compare the testing accuracy of applying the PMM from He et al. (2023) on $[0, 1]^{64}$ directly and applying Algorithm 1 with a target dimension $d'$. From Figure 2, the low-dimensional algorithm significantly improves the result for $\varepsilon > 1$. When $\varepsilon \leq 1$, direct PMM attains better accuracy. This is because when $\varepsilon n$ is too small, $(\varepsilon n)^{-1/d'}$ did not substantially reduce the error, so the advantage of low dimension has not been realized.

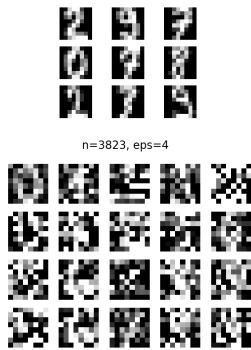

n=3823, eps=4

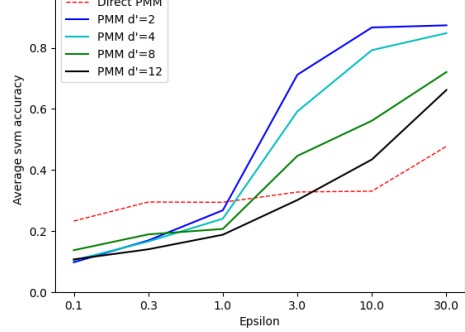

Figure 1: Original images (above) and synthetic images with 70% accuracy (below).

Figure 2: Testing accuracy for the SVM classifier trained on the synthetic datasets generated by Algorithm 1 with different $d'$ and $\varepsilon$.

## 6 CONCLUSION

In this paper, we provide a DP algorithm to generate synthetic data, which closely approximates the true data in the hypercube $[0, 1]^d$ under 1-Wasserstein distance. Moreover, when the true data lies in a $d'$-dimensional affine subspace, we improve the accuracy guarantees in (He et al., 2023) and circumvents the curse of dimensionality by generating a synthetic dataset close to the affine subspace.

It remains open to determine the optimal dependence on $d$ in the accuracy bound in Theorem 4.2 and whether the third term in (4.1) is needed. Our analysis of private PCA works without using the classical Davis-Kahan inequality that requires a spectral gap on the dataset. However, to approximate a dataset close to a line ($d' = 1$), additional assumptions are needed in our analysis to achieve the near-optimal accuracy rate, see Appendix D. It is an interesting problem to achieve an optimal rate without the dependence on $\sigma_1(\mathbf{M})$ when $d' = 1$.

Our Algorithm 1 only outputs synthetic data with a low-dimensional linear structure, and its analysis heavily relies on linear algebra tools. For original datasets from a $d'$-dimensional linear subspace, we improve the accuracy rate from $(\varepsilon n)^{-1/(d'+1)}$ in (Donhauser et al., 2023) to $(\varepsilon n)^{-1/d'}$. It is also interesting to provide algorithms with optimal accuracy rates for datasets from general low-dimensional manifolds beyond the linear setting.

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

# A    USEFUL DEFINITIONS AND LEMMAS

Differentially private algorithms have a useful property that their sequential composition is also differentially private (Dwork & Roth, 2014, Theorem 3.16).

**Lemma A.1** (Theorem 3.16 in (Dwork & Roth, 2014)). *Suppose $\mathcal{M}_i$ is $\varepsilon_i$-differentially private for $i = 1, \ldots, m$, then the sequential composition $x \mapsto (\mathcal{M}_1(x), \ldots, \mathcal{M}_m(x))$ is $\sum_{i=1}^m \varepsilon_i$-differentially private.*

Moreover, the following result about *adaptive composition* indicates that algorithms in a sequential composition may use the outputs in the previous steps:

**Lemma A.2** (Theorem 1 in (Dwork et al., 2006)). *Suppose a randomized algorithm $\mathcal{M}_1(x) : \Omega^n \to \mathcal{R}_1$ is $\varepsilon_1$-differentially private, and $\mathcal{M}_2(x, y) : \Omega^n \times \mathcal{R}_1 \to \mathcal{R}_2$ is $\varepsilon_2$-differentially private with respect to the first component for any fixed $y$. Then the sequential composition*

$$x \mapsto (\mathcal{M}_1(x), \mathcal{M}_2(x, \mathcal{M}_1(x)))$$

*is $(\varepsilon_1 + \varepsilon_2)$-differentially private.*

The formal definition of $p$-Wasserstein distance is given as follows:

**Definition A.3** ($p$-Wasserstein distance). *Consider a metric space $(\Omega, \rho)$. The $p$-Wasserstein distance (see e.g., (Villani, 2009) for more details) between two probability measures $\mu, \nu$ is defined as*

$$W_p(\mu, \nu) := \left( \inf_{\gamma \in \Gamma(\mu, \nu)} \int_{\Omega \times \Omega} \rho(x, y)^p \mathrm{d}\gamma(x, y) \right)^{1/p},$$

*where $\Gamma(\mu, \nu)$ is the set of all couplings of $\mu$ and $\nu$.*

When $p = 1$, the Kantorovich-Rubinstein duality (see, e.g., (Villani, 2009)) gives an equivalent representation of the 1-Wasserstein distance:

$$W_1(\mu, \nu) = \sup_{\mathrm{Lip}(f) \leq 1} \left( \int f \mathrm{d}\mu - \int f \mathrm{d}\nu \right),$$

where the supremum is taken over the set of all 1-Lipschitz functions on $\Omega$.

# B    MORE DETAILS ABOUT THE LOW-DIMENSIONAL SYNTHETIC DATA SUBROUTINES IN ALGORITHM 1

In this section, we include more details about the low-dimensional synthetic data subroutines in Algorithm 1. We first include two definitions used in He et al. (2023) below.

**Definition B.1** (Integer Laplacian distribution, Inusah & Kozubowski (2006)). *An* integer (or discrete) Laplacian distribution *with parameter $\sigma$ is a discrete distribution on $\mathbb{Z}$ with probability density function*

$$f(z) = \frac{1 - p_\sigma}{1 + p_\sigma} \exp\left( -|z| / \sigma \right), \quad z \in \mathbb{Z},$$

*where $p_\sigma = \exp(-1/\sigma)$. Thus a random variable $Z \sim \mathrm{Lap}_{\mathbb{Z}}(\sigma)$ is mean-zero and sub-exponential with variance $\mathrm{Var}(\mathrm{Z}) \leq 2\sigma^2$*

**Definition B.2** (Binary hierarchical partition, He et al. (2023)). *A binary hierarchical partition of a set $\Omega$ of depth $r$ is a family of subsets $\Omega_\theta$ indexed by $\theta \in \{0, 1\}^{\leq r}$, where*

$$\{0, 1\}^{\leq k} = \{0, 1\}^0 \sqcup \{0, 1\}^1 \sqcup \cdots \sqcup \{0, 1\}^k, \quad k = 0, 1, 2 \ldots,$$

*and such that $\Omega_\theta$ is partitioned into $\Omega_{\theta 0}$ and $\Omega_{\theta 1}$ for every $\theta \in \{0, 1\}^{\leq r-1}$. By convention, the cube $\{0, 1\}^0$ corresponds to $\emptyset$ and we write $\Omega_\emptyset = \Omega$.*

## B.1 PMM FOR $d' = 2$

A more detailed description of Algorithm 4 is as follows. For the region $\Omega'$, an $\ell_2$-ball of radius $R$, we first enlarge it into a hypercube $\Omega_{\mathrm{PMM}}$ of edge length $2R$ defined in Algorithm 4 inside the subspace $\widehat{\mathbf{V}}_{d'}$.

Next, for the hypercube $\Omega_{\mathrm{PMM}}$, we obtain a binary hierarchical partition $\{\Omega_\theta\}_{\theta \in \{0,1\}^{\leq r}}$ for $r = \lceil \log_2(\varepsilon n) \rceil$ by doing equal divisions of the hypercube recursively for $r$ rounds. Each round after the division, we count the data points in every new subregion $\Omega_\theta$ and add integer Laplacian noise to it.

Finally, a consistency step ensures the output is a well-defined probability measure. Here, the counts are considered to be *consistent* if they are non-negative and the counts of two smaller subregions at level $j$ can add up to the counts of the larger regions containing them at level $j-1$ for all $j \in [r]$.

## B.2 PSMM FOR $d' \geq 3$

The Private Signed Measure Mechanism (PSMM) introduced in He et al. (2023) generates a synthetic dataset $\mathbf{Y}$ in a compact domain $\Omega$ whose empirical measure $\mu_{\mathbf{Y}}$ is close to the empirical measure $\mu_{\mathbf{X}}$ of the original dataset $\mathbf{X}$ under the 1-Wasserstein distance.

PSMM runs in polynomial time, and the main steps are as follows. We first partition the domain $\Omega$ into $m$ disjoint subregions $\Omega_1, \ldots, \Omega_m$ and count the number of data points in each subregion. Then, we perturb the counts in each subregion with i.i.d. integer Laplacian noise. Based on the noisy counts, one can approximate $\mu_{\mathbf{X}}$ with a signed measure $\nu$ supported on $m$ points. Then, we find the closest probability measure $\hat{\nu}$ to the signed measure $\nu$ under the bounded Lipschitz distance by solving a linear programming problem.

We provide the main steps of PSMM in Algorithm 5. Details about the linear programming in the *synthetic probability measure* step can be found in (He et al., 2023). We apply PSMM from (He et al., 2023) when the metric space is an $\ell_2$-ball of radius $R$ inside $\widehat{\mathbf{V}}_{d'}$ and the following privacy and accuracy guarantees hold:

**Proposition B.3.** *The subroutine Algorithm 5 is $\varepsilon$-differentially private. And when $d' \geq 3$, with the input as the projected data $\widehat{\mathbf{X}}$ and the range $\Omega'$ with radius $R$ the algorithm has an accuracy bound*

$$\mathbb{E}W_1(\mu_{\widehat{\mathbf{X}}}, \mu_{\mathbf{X}'}) \lesssim \frac{R}{\sqrt{d'}}(\varepsilon n)^{-1/d'},$$

*where the expectation is conditioned on $R$.*

---

**Algorithm 5** PSMM Subroutine

---

**Input:** dataset $\widehat{\mathbf{X}} = (\widehat{X}_1, \ldots, \widehat{X}_n)$ in the region

$$\Omega' = \{a_1 \widehat{v}_1 + \cdots + a_{d'} \widehat{v}_{d'} \mid a_1^2 + \cdots + a_{d'}^2 \leq R^2\}.$$

**Integer lattice** Let $\delta = \sqrt{d/d'}(\varepsilon n)^{-1/d'}$. Find the lattice over the region:

$$L = \{a_1 \widehat{v}_1 + \cdots + a_{d'} \widehat{v}_{d'} \mid a_1^2 + \cdots + a_{d'}^2 \leq R^2, a_1, \ldots, a_{d'} \in \delta\mathbb{Z}\}.$$

**Counting** For any $v = a_1 \widehat{v}_1 + \cdots + a_{d'} \widehat{v}_{d'} \in L$, count the number

$$n_v = \left| \widehat{\mathbf{X}} \cap \{b_1 \widehat{v}_1 + \cdots + b_{d'} \widehat{v}_{d'} \mid b_i \in [a_i, a_i + \delta)\} \right|.$$

**Adding noise** Define a synthetic signed measure $\nu$ such that

$$\nu(\{v\}) = (n_v + \lambda_v)/n,$$

where $\lambda_v \sim \mathrm{Lap}_{\mathbb{Z}}(1/\varepsilon)$, $v \in L$ are i.i.d. random variables.

**Synthetic probability measure** Use linear programming and find the closest probability measure to $\nu$.

**Output:** Synthetic data corresponding to the probability measure.

---

**Remark B.4** (PMM vs PSMM for $d' \geq 2$). *For general $d' \geq 2$, PMM can still be applied, and the accuracy bound becomes $\mathbb{E}W_1(\mu_{\widehat{\mathbf{X}}}, \mu_{\mathbf{X}'}) \leq CR(\varepsilon n)^{-1/d'}$. Compared to (1.3), as $\mathbb{E}_\lambda R = \Theta(\sqrt{d})$, this accuracy bound is weaker by a factor of $\sqrt{d'}$. However, as shown in (He et al., 2023), PMM has a running time linear in $n$ and $d$, which is more computationally efficient than PSMM given in Algorithm 5 with running time polynomial in $n, d$.*

## C PROOFS

### C.1 PROOF OF THEOREM 2.1

*Proof.* Before applying the definition of differential privacy, we compute the entries of $\mathbf{M}$ explicitly. One can easily check that

$$\mathbf{M} = \frac{1}{n} \sum_{k=1}^{n} X_k X_k^{\mathsf{T}} - \frac{1}{n(n-1)} \sum_{k \neq \ell} X_k X_\ell^{\mathsf{T}}. \tag{C.1}$$

Now, if there are neighboring datasets $\mathbf{X}$ and $\mathbf{X}'$, suppose $X_k = (X_k^{(1)}, \ldots, X_k^{(d)})^{\mathsf{T}}$ is a column vector in $\mathbf{X}$ and $X_k' = (X_k'^{(1)}, \ldots, X_k'^{(d)})^{\mathsf{T}}$ is a column vector in $\mathbf{X}'$, and all other column vectors are the same. Let $\mathbf{M}$ and $\mathbf{M}'$ be the covariance matrix of $\mathbf{X}$ and $\mathbf{X}'$, respectively. Then we consider the density function ratio for the output of Algorithm 2 with input $\mathbf{X}$ and $\mathbf{X}'$:

$$\frac{\operatorname{den}_A(\widehat{\mathbf{M}} - \mathbf{M})}{\operatorname{den}_A(\widehat{\mathbf{M}} - \mathbf{M}')} = \prod_{i<j} \frac{\operatorname{den}_{\lambda_{ij}}((\widehat{\mathbf{M}} - \mathbf{M})_{ij})}{\operatorname{den}_{\lambda_{ij}}((\widehat{\mathbf{M}} - \mathbf{M}')_{ij})} \prod_{i=j} \frac{\operatorname{den}_{2\lambda_{ij}}((\widehat{\mathbf{M}} - \mathbf{M})_{ij})}{\operatorname{den}_{2\lambda_{ij}}((\widehat{\mathbf{M}} - \mathbf{M}')_{ij})}$$

$$= \prod_{i<j} \frac{\exp\left(-\frac{|(\widehat{\mathbf{M}}-\mathbf{M})_{ij}|}{\sigma}\right)}{\exp\left(-\frac{|(\widehat{\mathbf{M}}-\mathbf{M}')_{ij}|}{\sigma}\right)} \prod_i \frac{\exp\left(-\frac{|(\widehat{\mathbf{M}}-\mathbf{M})_{ii}|}{2\sigma}\right)}{\exp\left(-\frac{|(\widehat{\mathbf{M}}-\mathbf{M}')_{ii}|}{2\sigma}\right)}$$

$$\leq \exp\left(\sum_{i<j} \left|\mathbf{M}_{ij} - \mathbf{M}'_{ij}\right|/\sigma + \sum_i \left|\mathbf{M}_{ii} - \mathbf{M}'_{ii}\right|/(2\sigma)\right)$$

$$= \exp\left(\frac{1}{2\sigma} \sum_{i,j} \left|\mathbf{M}_{ij} - \mathbf{M}'_{ij}\right|\right).$$

As the datasets differs on only one data $X_k$, consider all entry containing $X_k$ in (C.1), we have

$$\left|\mathbf{M}_{ij} - \mathbf{M}'_{ij}\right| \leq \frac{1}{n}\left|X_k^{(i)} X_k^{(j)} - X_k'^{(i)} X_k'^{(j)}\right| + \frac{1}{n(n-1)} \sum_{\ell \neq k} \left|X_k^{(i)} - X_k'^{(i)}\right| X_\ell^{(j)}$$

$$+ \frac{1}{n(n-1)} \sum_{\ell \neq k} X_\ell^{(i)} \left|X_k^{(j)} - X_k'^{(j)}\right|$$

$$\leq \frac{2}{n} + \frac{2}{n(n-1)} \cdot 2(n-1) = \frac{6}{n}.$$

Therefore, substituting the result in the probability ratio implies

$$\frac{\operatorname{den}_A(\widehat{\mathbf{M}} - \mathbf{M})}{\operatorname{den}_A(\widehat{\mathbf{M}} - \mathbf{M}')} \leq \exp\left(\frac{1}{2\sigma} \cdot d^2 \cdot \frac{6}{n}\right) = \exp\left(\frac{3d^2}{\sigma n}\right),$$

and when $\sigma = \frac{3d^2}{\varepsilon n}$, Algorithm 2 is $\varepsilon$-differentially private. $\qquad \square$

### C.2 PROOF OF LEMMA 2.2

*Proof.* Let $\widehat{v}_1, \ldots, \widehat{v}_d$ be a set of orthonormal eigenvectors for $\widehat{\mathbf{M}}$ with the corresponding eigenvalues $\widehat{\sigma}_1, \ldots, \widehat{\sigma}_d$. Define four matrices whose column vectors are eigenvectors:

$$\mathbf{V} = [v_1, \ldots, v_d], \qquad \widehat{\mathbf{V}} = [\widehat{v}_1, \ldots, \widehat{v}_d],$$
$$\mathbf{V}_{d'} = [v_1, \ldots, v_{d'}], \qquad \widehat{\mathbf{V}}_{d'} = [\widehat{v}_1, \ldots, \widehat{v}_{d'}].$$

By orthogonality, the following identities hold:

$$\sum_{i=1}^{d} \|v_i^{\mathsf{T}} \mathbf{X}\|_2^2 = \sum_{i=1}^{d} \|\widehat{v}_i^{\mathsf{T}} \mathbf{X}\|_2^2 = \|\mathbf{X}\|_F^2.$$

$$\sum_{i>d'} \|v_i^{\mathsf{T}} \mathbf{X}\|_2^2 = \|\mathbf{X} - \mathbf{V}_{d'} \mathbf{V}_{d'}^{\mathsf{T}} \mathbf{X}\|_F^2.$$

$$\sum_{i>d'} \|\widehat{v}_i^{\mathsf{T}} \mathbf{X}\|_2^2 = \|\mathbf{X} - \widehat{\mathbf{V}}_{d'} \widehat{\mathbf{V}}_{d'}^{\mathsf{T}} \mathbf{X}\|_F^2.$$

Separating the top $d'$ eigenvectors from the rest, we obtain

$$\sum_{i \leq d'} \|v_i^{\mathsf{T}} \mathbf{X}\|_2^2 + \|\mathbf{X} - \mathbf{V}_{d'} \mathbf{V}_{d'}^{\mathsf{T}} \mathbf{X}\|_F^2 = \sum_{i \leq d'} \|\widehat{v}_i^{\mathsf{T}} \mathbf{X}\|_2^2 + \|\mathbf{X} - \widehat{\mathbf{V}}_{d'} \widehat{\mathbf{V}}_{d'}^{\mathsf{T}} \mathbf{X}\|_F^2.$$

Therefore

$$\begin{aligned}
\|\mathbf{X} - \widehat{\mathbf{V}}_{d'} \widehat{\mathbf{V}}_{d'}^{\mathsf{T}} \mathbf{X}\|_F^2 &= \sum_{i \leq d'} \|v_i^{\mathsf{T}} \mathbf{X}\|_2^2 - \sum_{i \leq d'} \|\widehat{v}_i^{\mathsf{T}} \mathbf{X}\|_2^2 + \|\mathbf{X} - \mathbf{V}_{d'} \mathbf{V}_{d'}^{\mathsf{T}} \mathbf{X}\|_F^2 \\
&= n \sum_{i \leq d'} \sigma_i - n \sum_{i \leq d'} \widehat{v}_i^{\mathsf{T}} \mathbf{M} \widehat{v}_i + n \sum_{i>d'} \sigma_i \\
&= n \sum_{i \leq d'} \sigma_i - n \sum_{i \leq d'} \widehat{v}_i^{\mathsf{T}} (\widehat{\mathbf{M}} - \mathbf{A}) \widehat{v}_i + n \sum_{i>d'} \sigma_i \\
&= n \sum_{i \leq d'} (\sigma_i - \widehat{\sigma}_i) + n \operatorname{tr}(\mathbf{A} \widehat{\mathbf{V}}_{d'} \widehat{\mathbf{V}}_{d'}^{\mathsf{T}}) + n \sum_{i>d'} \sigma_i. \quad (\text{C.2})
\end{aligned}$$

By Weyl's inequality, for $i \leq d'$,

$$|\sigma_i - \widehat{\sigma}_i| \leq \|\mathbf{A}\|. \quad (\text{C.3})$$

By von Neumann's trace inequality,

$$\operatorname{tr}(A \widehat{\mathbf{V}}_{d'} \widehat{\mathbf{V}}_{d'}^{\mathsf{T}}) \leq \sum_{i=1}^{d'} \sigma_i(\mathbf{A}). \quad (\text{C.4})$$

From (C.2), (C.3), and (C.4),

$$\frac{1}{n} \|\mathbf{X} - \widehat{\mathbf{V}}_{d'} \widehat{\mathbf{V}}_{d'}^{\mathsf{T}} \mathbf{X}\|_F^2 \leq \sum_{i>d'} \sigma_i + d' \|\mathbf{A}\| + \sum_{i=1}^{d'} \sigma_i(\mathbf{A}) \leq \sum_{i>d'} \sigma_i + 2d' \|\mathbf{A}\|.$$

Let $Y_i$ be the $i$-th column of $\mathbf{Y}$. We have

$$W_2^2(\mu_{\mathbf{X}}, \mu_{\mathbf{Y}}) \leq \frac{1}{n} \sum_{i=1}^{n} \|X_i - Y_i\|_2^2 = \frac{1}{n} \|\mathbf{X} - \mathbf{Y}\|_F^2.$$

Therefore (2.1) holds. □

### C.3 PROOF OF THEOREM 2.3

*Proof.* For the true covariance matrix $\mathbf{M}$, consider its SVD

$$\mathbf{M} = \frac{1}{n-1} \sum_{i=1}^{n} (X_i - \overline{X})(X_i - \overline{X})^{\mathsf{T}} = \sum_{j=1}^{d} \sigma_j v_j v_j^{\mathsf{T}}, \quad (\text{C.5})$$

where $\sigma_1 \geq \sigma_2 \geq \cdots \geq \sigma_d$ are the singular values and $v_1 \ldots v_d$ are corresponding orthonormal eigenvectors. Moreover, define two $d \times d'$ matrices

$$\mathbf{V}_{d'} = [v_1, \ldots, v_{d'}], \qquad \widehat{\mathbf{V}}_{d'} = [\widehat{v}_1, \ldots, \widehat{v}_{d'}].$$

Then the matrix $\widehat{\mathbf{V}}_{d'}\widehat{\mathbf{V}}_{d'}^{\mathsf{T}}$ is a projection onto the subspace spanned by the principal components $\widehat{v}_1, \ldots, \widehat{v}_{d'}$.

In Algorithm 3, for any data $X_i$ we first shift it to $X_i - \overline{X} - \lambda$ and then project it to $\widehat{\mathbf{V}}_{d'}\widehat{\mathbf{V}}_{d'}^{\mathsf{T}}(X_i - \overline{X} - \lambda)$. Therefore

$$
\begin{aligned}
\left\| X_i - \overline{X} - \widehat{\mathbf{V}}_{d'}\widehat{\mathbf{V}}_{d'}^{\mathsf{T}}(X_i - \overline{X} - \lambda) \right\|_\infty &\leq \left\| X_i - \overline{X} - \widehat{\mathbf{V}}_{d'}\widehat{\mathbf{V}}_{d'}^{\mathsf{T}}(X_i - \overline{X}) \right\|_\infty + \left\| \widehat{\mathbf{V}}_{d'}\widehat{\mathbf{V}}_{d'}^{\mathsf{T}}\lambda \right\|_\infty \\
&\leq \left\| X_i - \overline{X} - \widehat{\mathbf{V}}_{d'}\widehat{\mathbf{V}}_{d'}^{\mathsf{T}}(X_i - \overline{X}) \right\|_2 + \|\lambda\|_2 .
\end{aligned}
$$

Let $Z_i$ denote $X_i - \overline{X}$ and $\mathbf{Z} = [Z_1, \ldots, Z_n]$. Then

$$
\frac{1}{n}\mathbf{Z}\mathbf{Z}^{\mathsf{T}} = \frac{n-1}{n}\mathbf{M}.
$$

With Lemma 2.2, by definition of the Wasserstein distance, we have

$$
\begin{aligned}
W_2^2(\mu_{\mathbf{X}-\overline{X}\mathbf{1}^{\mathsf{T}}}, \mu_{\widehat{\mathbf{X}}}) &= \frac{1}{n}\sum_{i=1}^n \left\| X_i - \overline{X} - \widehat{\mathbf{V}}_{d'}\widehat{\mathbf{V}}_{d'}^{\mathsf{T}}(X_i - \overline{X} - \lambda) \right\|_\infty^2 \\
&\leq \frac{2}{n}\sum_{i=1}^n \left\| X_i - \overline{X} - \widehat{\mathbf{V}}_{d'}\widehat{\mathbf{V}}_{d'}^{\mathsf{T}}(X_i - \overline{X}) \right\|_2^2 + 2\|\lambda\|_2^2 \\
&= \frac{2}{n}\|\mathbf{Z} - \widehat{\mathbf{V}}_{d'}\widehat{\mathbf{V}}_{d'}^{\mathsf{T}}\mathbf{Z}\|_F^2 + 2\|\lambda\|_2^2 \\
&\leq 2\sum_{i=d'}^n \sigma_i(\mathbf{M}) + 4d'\|\mathbf{A}\| + 2\|\lambda\|_2^2 . \quad\quad\quad (C.6)
\end{aligned}
$$

Since $\lambda = (\lambda_1, \ldots, \lambda_d)$ is a Laplacian random vector with i.i.d. $\mathrm{Lap}(1/(\varepsilon n))$ entries,

$$
\mathbb{E}\|\lambda\|_2^2 = \sum_{j=1}^d \mathbb{E}\big|\lambda_j\big|^2 = \frac{2d}{\varepsilon^2 n^2}. \quad\quad\quad (C.7)
$$

Furthermore, in Algorithm 2, $A$ is a symmetric random matrix with independent Laplacian random variables on and above its diagonal. Thus, we have the tail bound for its norm (Dai et al., 2022, Theorem 1.1)

$$
\mathbb{P}\left\{ \|\mathbf{A}\| \geq \sigma(C\sqrt{d}+t) \right\} \leq C_0 \exp(-C_1 \min(t^2/4, t/2)). \quad\quad\quad (C.8)
$$

And we can further compute the expectation bound for $\|\mathbf{A}\|$ from (C.8) with the choice of $\sigma = \frac{3d^2}{\varepsilon n}$,

$$
\mathbb{E}\|\mathbf{A}\| \leq C\sigma\sqrt{d} + \int_0^\infty C_0 \exp\left(-C_1 \min\left(\frac{t^2}{4\sigma^2}, \frac{t}{2\sigma}\right)\right)\mathrm{d}t \lesssim \frac{d^{2.5}}{\varepsilon n}.
$$

Combining the two bounds above and (C.6), as the 1-Wasserstein distance is bounded by the 2-Wasserstein distance and inequality $\sqrt{x+y} \leq \sqrt{x} + \sqrt{y}$ holds for all $x, y \geq 0$,

$$
\begin{aligned}
\mathbb{E}W_1(\mu_{\mathbf{X}-\overline{X}\mathbf{1}^{\mathsf{T}}}, \mu_{\widehat{\mathbf{X}}}) &\leq \left(\mathbb{E}W_2^2(\mu_{\mathbf{X}-\overline{X}\mathbf{1}^{\mathsf{T}}}, \mu_{\widehat{\mathbf{X}}})\right)^{1/2} \\
&\leq \sqrt{2\sum_{i>d'}\sigma_i(\mathbf{M})} + \sqrt{4d'\mathbb{E}\|\mathbf{A}\|} + \sqrt{2\mathbb{E}\|\lambda\|_2^2} \\
&\leq \sqrt{2\sum_{i>d'}\sigma_i(\mathbf{M})} + \sqrt{\frac{Cd'd^{2.5}}{\varepsilon n}}.
\end{aligned}
$$

$\square$

## C.4 Proof of Proposition 3.1

*Proof.* The privacy guarantee follows from (He et al., 2023, Theorem 1.1). For accuracy, note that the region $\Omega'$ is a subregion of a $d'$-dimensional ball. Algorithm 4 enlarges the region to a $d'$-dimensional hypercube with side length $2R$. By re-scaling the size of the hypercube and applying (He et al., 2023, Corollary 4.4), we obtain the accuracy bound. □

## C.5 Proof of Theorem 4.1

*Proof.* We can decompose Algorithm 1 into the following steps:

1. $\mathcal{M}_1(\mathbf{X}) = \widehat{\mathbf{M}}$ is to compute the private covariance matrix with Algorithm 2.

2. $\mathcal{M}_2(\mathbf{X}) = \overline{X} + \lambda$ is to compute the private sample mean.

3. $\mathcal{M}_3(\mathbf{X}, y, \Sigma)$ for fixed $y$ and $\Sigma$, is to project the shifted data $\{X_i - y\}_{i=1}^n$ to the first $d'$ principal components of $\Sigma$ and apply a certain differentially private subroutine (we choose $y$ and $\Sigma$ as the output of $\mathcal{M}_2$ and $\mathcal{M}_1$, respectively). This step outputs synthetic data $\mathbf{X}' = (X_1', \ldots, X_m')$ on a linear subspace.

4. $\mathcal{M}_4(\mathbf{X}, \mathbf{X}')$ is to shift the dataset to $\{X_i' + \overline{X}_{\text{priv}}\}_{i=1}^m$.

5. Metric projection.

It suffices to show that the data before metric projection has already been differentially private. We will need to apply Lemma A.2 several times.

With respect to the input $\mathbf{X}$ while fixing other input variables, we know that $\mathcal{M}_1, \mathcal{M}_2, \mathcal{M}_3, \mathcal{M}_4$ are all $\varepsilon/4$-differentially private. Therefore, by using Lemma A.2 iteratively, the composition algorithm

$$\mathcal{M}_4(\mathbf{X}, \mathcal{M}_3(\mathbf{X}, \mathcal{M}_2(\mathbf{X}), \mathcal{M}_1(\mathbf{X})))$$

satisfies $\varepsilon$-differential privacy. Hence Algorithm 1 is $\varepsilon$-differentially private. □

## C.6 Proof of Theorem 4.2

*Proof.* Similar to privacy analysis, we will decompose the algorithm into several steps. Suppose that

1. $\mathbf{X} - (\overline{X} + \lambda)\mathbf{1}^\mathsf{T}$ denotes the shifted data $\{X_i - \overline{X} - \lambda\}_{i=1}^n$;

2. $\widehat{\mathbf{X}}$ is the data after projection to the private linear subspace;

3. $\mathbf{X}'$ is the output of the synthetic data subroutine in Section 3;

4. $\mathbf{X}'' = \mathbf{X}' + (\overline{X} + \lambda')\mathbf{1}^\mathsf{T}$ denotes the data shifted back;

5. $\mathcal{M}(\mathbf{X})$ is the data after metric projection, which is the output of the whole algorithm.

For the metric projection step, by Proposition 3.2, we have that

$$
\begin{aligned}
W_1(\mu_{\mathbf{X}}, \mu_{\mathcal{M}(\mathbf{X})}) &\le W_1(\mu_{\mathbf{X}}, \mu_{\mathbf{X}''}) + W_1(\mu_{\mathbf{X}''}, \mu_{\mathcal{M}(\mathbf{X})}) \\
&\le 2W_1(\mu_{\mathbf{X}}, \mu_{\mathbf{X}''}).
\end{aligned}
\tag{C.9}
$$

Moreover, applying the triangle inequality of Wasserstein distance to the other steps of the algorithm, we have

$$
\begin{aligned}
W_1(\mu_{\mathbf{X}}, \mu_{\mathbf{X}''}) &= W_1(\mu_{\mathbf{X}-\overline{X}\mathbf{1}^\mathsf{T}}, \mu_{\mathbf{X}'+\lambda'\mathbf{1}^\mathsf{T}}) \\
&\le W_1(\mu_{\mathbf{X}-\overline{X}\mathbf{1}^\mathsf{T}}, \mu_{\widehat{\mathbf{X}}}) + W_1(\mu_{\widehat{\mathbf{X}}}, \mu_{\mathbf{X}'}) + W_1(\mu_{\mathbf{X}'}, \mu_{\mathbf{X}'+\lambda'}) \\
&\le W_1(\mu_{\mathbf{X}-\overline{X}\mathbf{1}^\mathsf{T}}, \mu_{\widehat{\mathbf{X}}}) + W_1(\mu_{\widehat{\mathbf{X}}}, \mu_{\mathbf{X}'}) + \left\|\lambda'\right\|_\infty.
\end{aligned}
\tag{C.10}
$$

Note that $W_1(\mu_{\mathbf{X}-\overline{X}\mathbf{1}^\intercal}, \mu_{\widehat{\mathbf{X}}})$ is the projection error we bound in Theorem 2.3, and $W_1(\mu_{\widehat{\mathbf{X}}}, \mu_{\mathbf{X}'})$ is treated in the accuracy analysis of subroutines in Section 3. Moreover, we have

$$
\begin{aligned}
\mathbb{E}W_1(\mu_{\widehat{\mathbf{X}}}, \mu_{\mathbf{X}'}) &= \mathbb{E}_R \mathbb{E}_{\mathbf{X}'} W_1(\mu_{\widehat{\mathbf{X}}}, \mu_{\mathbf{X}'}) \\
&\leq \mathbb{E}_R \frac{CR}{\sqrt{d'}} (\varepsilon n)^{-1/d'} \\
&\leq \frac{C(2\sqrt{d} + \mathbb{E}\|\lambda\|_2)}{\sqrt{d'}} (\varepsilon n)^{-1/d'} \\
&\lesssim \sqrt{\frac{d}{d'}} (\varepsilon n)^{-1/d'}.
\end{aligned}
$$

Here in the last step we use $\mathbb{E}\|\lambda\|_2 \leq \frac{C\sqrt{d}}{\varepsilon n}$ in (C.7). Since $\lambda'$ is a sub-exponential random vector, the following bound also holds for some absolute constant $C > 0$:

$$
\mathbb{E}\|\lambda'\|_\infty \leq \frac{C \log d}{\varepsilon n}. \tag{C.11}
$$

Hence

$$
\begin{aligned}
&\mathbb{E}W_1(\mu_{\mathbf{X}}, \mu_{\mathcal{M}(\mathbf{X})}) \\
&\leq 2\mathbb{E}W_1(\mu_{\mathbf{X}}, \mu_{\mathbf{X}'+(\overline{X}+\lambda')\mathbf{1}^\intercal}) \\
&\leq 2\mathbb{E}W_1(\mu_{\mathbf{X}-\overline{X}\mathbf{1}^\intercal}, \mu_{\widehat{\mathbf{X}}}) + 2\mathbb{E}W_1(\mu_{\widehat{\mathbf{X}}}, \mu_{\mathbf{X}'}) + 2\mathbb{E}\|\lambda'\|_\infty \\
&\leq 2\sqrt{2\sum_{i>d'} \sigma_i(\mathbf{M})} + 2\sqrt{\frac{Cd'd^{2.5}}{\varepsilon n}} + 2C\sqrt{\frac{d}{d'}}(\varepsilon n)^{-1/d'} + \frac{2C\log d}{\varepsilon n} \tag{C.12} \\
&\lesssim \sqrt{\sum_{i>d'}\sigma_i(\mathbf{M})} + \sqrt{\frac{d}{d'}}(\varepsilon n)^{-1/d'} + \sqrt{\frac{d'd^{2.5}}{\varepsilon n}},
\end{aligned}
$$

where the first inequality is from (C.9), the second inequality is from (C.10), and the third inequality is due to Theorem 2.3, Proposition 3.1, and Proposition B.3. $\qquad\square$

## C.7 PROOF OF PROPOSITION 3.2

*Proof.* For the function $f$ defined in Algorithm 1, we know $f(x)$ is the closest real number to $x$ in the region $[0,1]$ for any $x \in \mathbb{R}$. Furthermore, if $v \in \mathbb{R}^d$ is a vector, then $f(v)$ is the closest vector to $v$ in $[0,1]^d$ with respect to $\|\cdot\|_\infty$. Thus $f : \mathbb{R}^d \to [0,1]^d$ is indeed a metric projection to $[0,1]^d$.

We first assume that the synthetic data $\mathbf{X}''$ also has size $n$. Then for any column vector $X_i''$, we know that $Y_i = f(X_i'')$ is its closest vector in $[0,1]^d$ under the $\ell^\infty$ metric. For the data $X_1, X_2, \ldots, X_n$, suppose that the solution to the optimal transportation problem for $W_1(\mu_{\mathbf{X}}, \mu_{\mathbf{X}''})$ is to match $X_{\tau(i)}$ with $X_i''$, where $\tau$ is a permutation on $[n]$. Then

$$
W_1(\mu_{\mathbf{Y}}, \mu_{\mathbf{X}''}) \leq \frac{1}{n}\sum_{i=1}^n \|Y_i - X_i''\|_\infty \leq \frac{1}{n}\sum_{i=1}^n \|X_{\tau(i)} - X_i''\|_\infty = W_1(\mu_{\mathbf{X}}, \mu_{\mathbf{X}''}).
$$

In general, if the synthetic dataset has $m$ data points and $m \neq n$, we can split the points and regard both the true dataset and synthetic dataset as of size $mn$, then it's easy to check that the inequality still holds.

The expectation bound follows from (C.10) and (C.12). $\qquad\square$

## C.8 PROOF OF PROPOSITION B.3

*Proof.* The proposition is a direct corollary to the result in (He et al., 2023). The size of the scaled integer lattice $\delta\mathbb{Z}$ in the unit $d$-dimensional ball of radius $R$ is bounded by $(\frac{C}{\delta R})^d$ for an absolute

constant $C > 0$ (see, for example, (Feige & Ofek, 2005, Claim 2.9) and (Boedihardjo et al., 2022a, Proposition 3.7)). Then, the number of subregions in Algorithm 5 is bounded by

$$|L| \le \left( \frac{R}{\sqrt{d'}} \cdot \frac{C}{\delta} \right)^{d'}.$$

By (He et al., 2023, Theorem 3.6), we have

$$\mathbb{E}W_1(\mu_{\widehat{\mathbf{X}}}, \mu_{\mathbf{X}'}) \le \delta + \frac{2}{\varepsilon n} \left( \frac{R}{\sqrt{d'}} \cdot \frac{C}{\delta} \right)^{d'} \cdot \frac{1}{d'} \left( \left( \frac{R}{\sqrt{d'}} \cdot \frac{C}{\delta} \right)^{d'} \right)^{-\frac{1}{d'}}.$$

Taking $\delta = \frac{CR}{\sqrt{d'}}(\varepsilon n)^{-\frac{1}{d'}}$ concludes the proof. $\qquad\qquad\square$

# D NEAR-OPTIMAL ACCURACY BOUND WITH ADDITIONAL ASSUMPTIONS WHEN $d' = 1$

Our Theorem 4.2 is not applicable to the case $d' = 1$ because the projection error in Theorem 2.3 only has bound $O((\varepsilon n)^{-\frac{1}{2}})$, which does not match with the optimal synthetic data accuracy bound in (Boedihardjo et al., 2022b; He et al., 2023). We are able to improve the accuracy bound with an additional dependence on $\sigma_1(\mathbf{M})$ as follows:

**Theorem D.1.** *When $d' = 1$, consider Algorithm 1 with input data $\mathbf{X}$, output data $\mathbf{Y}$, and the subroutine PMM in Algorithm 4. Let $\mathbf{M}$ be the covariance matrix defines as (1.2). Assume $\sigma_1(\mathbf{M}) > 0$, then*

$$\mathbb{E}W_1(\mu_{\mathbf{X}}, \mu_{\mathbf{Y}}) \lesssim \sqrt{\sum_{i>1} \sigma_i(\mathbf{M})} + \frac{d^3}{\sqrt{\sigma_1(\mathbf{M})}\varepsilon n} + \frac{\sqrt{d}\log^2(\varepsilon n)}{\varepsilon n}.$$

We start with the following lemma based on the Davis-Kahan theorem (Yu et al., 2015).

**Lemma D.2.** *Let $\mathbf{X}$ be a $d \times n$ matrix and $\mathbf{A}$ be an $d \times d$ Hermitian matrix. Let $\mathbf{M} = \frac{1}{n}\mathbf{X}\mathbf{X}^{\mathsf{T}}$, with the SVD*

$$\mathbf{M} = \sum_{j=1}^{d} \sigma_j v_j v_j^{\mathsf{T}},$$

*where $\sigma_1 \ge \sigma_2 \ge \cdots \ge \sigma_d$ are the singular values of $\mathbf{M}$ and $v_1, \ldots, v_d$ are corresponding orthonormal eigenvectors. Let $\widehat{\mathbf{M}} = \frac{1}{n}\mathbf{X}\mathbf{X}^{\mathsf{T}} + \mathbf{A}$ with orthonormal eigenvectors $\widehat{v}_1, \ldots, \widehat{v}_d$, where $\widehat{v}_1$ corresponds to the top singular value of $\widehat{\mathbf{M}}$. When there exists a spectral gap $\sigma_1 - \sigma_2 = \delta > 0$, we have*

$$\frac{1}{n}\|\mathbf{X} - \widehat{v}_1\widehat{v}_1^{\mathsf{T}}\mathbf{X}\|_F^2 \le 2\sum_{i>d'} \sigma_i + \frac{8d'^2}{n\delta^2}\|\mathbf{A}\|^2\|\mathbf{X}\|_F^2.$$

*Proof.* We have that

$$\begin{aligned}
\frac{1}{n}\|\mathbf{X} - \widehat{v}_1\widehat{v}_1^{\mathsf{T}}\mathbf{X}\|_F^2 &= \frac{1}{n}\|\mathbf{X} - v_1 v_1^{\mathsf{T}}\mathbf{X} + v_1 v_1^{\mathsf{T}}\mathbf{X} - \widehat{v}_1\widehat{v}_1^{\mathsf{T}}\mathbf{X}\|_F^2 \\
&\le \frac{2}{n}\left( \|\mathbf{X} - v_1 v_1^{\mathsf{T}}\mathbf{X}\|_F^2 + \|v_1 v_1^{\mathsf{T}}\mathbf{X} - \widehat{v}_1\widehat{v}_1^{\mathsf{T}}\mathbf{X}\|_F^2 \right) \\
&= 2\sum_{i>d'} \sigma_i + \frac{2}{n}\left\| \left( v_1 v_1^{\mathsf{T}} - \widehat{v}_1\widehat{v}_1^{\mathsf{T}} \right)\mathbf{X} \right\|_F^2 \\
&\le 2\sum_{i>d'} \sigma_i + \frac{2}{n}\left\| v_1 v_1^{\mathsf{T}} - \widehat{v}_1\widehat{v}_1^{\mathsf{T}} \right\|^2 \|\mathbf{X}\|_F^2. \qquad\qquad (\text{D.1})
\end{aligned}$$

To bound the operator norm distance between the two projections, we will need the Davis-Kahan Theorem in the perturbation theory. For the angle $\Theta(v_1, \widehat{v}_1)$ between the vectors $v_1$ and $\widehat{v}_1$, applying (Yu et al., 2015, Corollary 1), we have

$$\left\| v_1 v_1^{\mathsf{T}} - \widehat{v}_1\widehat{v}_1^{\mathsf{T}} \right\| = \sin\Theta(v_1, \widehat{v}_1) \le \frac{2\|\mathbf{M} - \widehat{\mathbf{M}}\|}{\sigma_1 - \sigma_2} \le \frac{2\|\mathbf{A}\|}{\delta}.$$

Therefore, when the spectral gap exists ($\delta > 0$),

$$\frac{1}{n}\|\mathbf{X} - \widehat{v}_1\widehat{v}_1^\mathsf{T}\mathbf{X}\|_F^2 \le 2\sum_{i>d'}\sigma_i + \frac{8}{n\delta^2}\|\mathbf{A}\|^2\|\mathbf{X}\|_F^2.$$

$\square$

Compared to Lemma 2.2, with the extra spectral gap assumption, the dependence on $\mathbf{A}$ in the upper bound changes from $\|\mathbf{A}\|$ to $\|\mathbf{A}\|^2$. A similar phenomenon, called global and local bounds, was observed in (Reiss & Wahl, 2020, Proposition 2.2). With Lemma D.2, we are able to improve the accuracy rate for the noisy projection step as follows.

**Theorem D.3.** *When $d' = 1$, assume that $\sigma_1(\mathbf{M}) = \|\mathbf{M}\| > 0$. For the output $\widehat{\mathbf{X}}$ in Algorithm 3, we have*

$$\mathbb{E}W_1(\mu_{\mathbf{X}-\overline{X}\mathbf{1}^\mathsf{T}}, \mu_{\widehat{\mathbf{X}}}) \le \left(\mathbb{E}W_2^2(\mu_{\mathbf{X}-\overline{X}\mathbf{1}^\mathsf{T}}, \mu_{\widehat{\mathbf{X}}})\right)^{1/2} \lesssim \sqrt{\sum_{i>1}\sigma_i} + \frac{d^3}{\sqrt{\sigma_1}\varepsilon n},$$

*where $\sigma_1 \ge \cdots \ge \sigma_d \ge 0$ are singular values of $\mathbf{M}$.*

*Proof.* Similar to the proof of Theorem 2.3, we can define $Z_i = X_i - \overline{X}$ and deduce that

$$\frac{1}{n}\mathbf{Z}\mathbf{Z}^\mathsf{T} = \frac{n-1}{n}\mathbf{M},$$
$$\frac{1}{n}\|\mathbf{Z}\|_F^2 = \frac{n-1}{n}\operatorname{tr}(\mathbf{M}),$$

and

$$W_2^2(\mu_{\mathbf{X}-\overline{X}\mathbf{1}^\mathsf{T}}, \mu_{\widehat{\mathbf{X}}}) = \frac{2}{n}\|\mathbf{Z} - \widehat{v}_1\widehat{v}_1^\mathsf{T}\mathbf{Z}\|_F^2 + 2\|\lambda\|_2^2.$$

By the inequality $\sqrt{x+y} \le \sqrt{x} + \sqrt{y}$ for $x, y \ge 0$,

$$\mathbb{E}W_1(\mu_{\mathbf{X}-\overline{X}\mathbf{1}^\mathsf{T}}, \mu_{\widehat{\mathbf{X}}}) \le \mathbb{E}\left[\frac{2}{n}\|\mathbf{Z} - \widehat{v}_1\widehat{v}_1^\mathsf{T}\mathbf{Z}\|_F^2\right]^{1/2} + \sqrt{2}\mathbb{E}\|\lambda\|_2.$$

Let $\delta = \sigma_1 - \sigma_2$. Next, we will discuss two cases for the value of $\delta$.

**Case 1:** When $\delta = \sigma_1 - \sigma_2 \le \frac{1}{2}\sigma_1$, we have $\sigma_1 \le 2\sigma_2$ and

$$\operatorname{tr}(\mathbf{M}) = \sigma_1 + \cdots + \sigma_d \le 3\sum_{i>1}\sigma_i.$$

As any projection map has spectral norm 1, we have $\|v_1v_1^\mathsf{T} - \widehat{v}_1\widehat{v}_1^\mathsf{T}\| \le 2$. Applying (D.1), we have

$$\frac{1}{n}\|\mathbf{Z} - \widehat{v}_1\widehat{v}_1^\mathsf{T}\mathbf{Z}\|_F^2 \le 2\sum_{i>1}\sigma_i + \frac{2}{n}\left\|v_1v_1^\mathsf{T} - \widehat{v}_1\widehat{v}_1^\mathsf{T}\right\|^2\|\mathbf{Z}\|_F^2$$
$$\le 2\sum_{i>1}\sigma_i + \frac{8}{n}\|\mathbf{Z}\|_F^2$$
$$\le 2\sum_{i>1}\sigma_i + 8\operatorname{tr}(\mathbf{M})$$
$$\le 26\sum_{i>1}\sigma_i.$$

Hence

$$\mathbb{E}W_1(\mu_{\mathbf{X}-\overline{X}\mathbf{1}^\mathsf{T}}, \mu_{\widehat{\mathbf{X}}}) \lesssim \sqrt{\sum_{i>1}\sigma_i} + \mathbb{E}\|\lambda\|_2 \lesssim \sqrt{\sum_{i>1}\sigma_i} + \frac{\sqrt{d}}{\varepsilon n}. \tag{D.2}$$

**Case 2:** When $\delta \ge \frac{1}{2}\sigma_1$, we have

$$\operatorname{tr}(\mathbf{M}) \le d\sigma_1 \le \frac{4d\delta^2}{\sigma_1}.$$

For any fixed $\delta$, by Lemma D.2,

$$\frac{1}{n}\|\mathbf{Z} - \widehat{v}_1\widehat{v}_1^{\mathsf{T}}\mathbf{Z}\|_F^2 \leq 2\sum_{i>1}\sigma_i + \frac{8}{n\delta^2}\|\mathbf{A}\|^2\|\mathbf{Z}\|_F^2$$

$$\leq 2\sum_{i>1}\sigma_i + \frac{8}{\delta^2}\|\mathbf{A}\|^2\operatorname{tr}(\mathbf{M})$$

$$\leq 2\sum_{i>1}\sigma_i + \frac{32d}{\sigma_1}\|\mathbf{A}\|^2.$$

So we have the Wasserstein distance bound

$$\mathbb{E}W_1(\mu_{\mathbf{X}-\overline{X}\mathbf{1}^{\mathsf{T}}}, \mu_{\widehat{\mathbf{X}}}) \leq \sqrt{2\sum_{i>1}\sigma_i} + \sqrt{\frac{32d}{\sigma_1}}\mathbb{E}\|\mathbf{A}\| + \sqrt{2}\mathbb{E}\|\lambda\|_2$$

$$\leq \sqrt{2\sum_{i>1}\sigma_i} + \sqrt{\frac{32d}{\sigma_1}}\frac{d^{2.5}}{\varepsilon n} + \frac{\sqrt{2d}}{\varepsilon n}$$

$$\leq \sqrt{2\sum_{i>1}\sigma_i} + \frac{Cd^3}{\sqrt{\sigma_1}\varepsilon n}. \tag{D.3}$$

From (C.5),

$$\sigma_1 = \|M\| \leq \|M\|_F \leq \frac{n}{n-1}d \leq 2d.$$

Combining the two cases (D.2) and (D.3), we deduce the result. □

*Proof of Theorem D.1.* Following the steps in the proof of Theorem 2.3, we obtain

$$\mathbb{E}W_1(\mu_{\mathbf{X}}, \mu_{\mathcal{M}(\mathbf{X})}) \leq 2\mathbb{E}W_1(\mu_{\mathbf{X}}, \mu_{\mathbf{X}'+(\overline{X}+\lambda')\mathbf{1}^{\mathsf{T}}})$$

$$\leq 2\mathbb{E}W_1(\mu_{\mathbf{X}-\overline{X}\mathbf{1}^{\mathsf{T}}}, \mu_{\widehat{\mathbf{X}}}) + 2\mathbb{E}W_1(\mu_{\widehat{\mathbf{X}}}, \mu_{\mathbf{X}'}) + 2\mathbb{E}\|\lambda'\|_\infty$$

$$\lesssim \sqrt{\sum_{i>1}\sigma_i} + \frac{d'd^3}{\sqrt{\sigma_1}\varepsilon n} + \frac{\sqrt{d}\log^2(\varepsilon n)}{\varepsilon n} + \frac{2C\log d}{\varepsilon n}$$

$$\lesssim \sqrt{\sum_{i>1}\sigma_i} + \frac{d'd^3}{\sqrt{\sigma_1}\varepsilon n} + \frac{\sqrt{d}\log^2(\varepsilon n)}{\varepsilon n},$$

where for the second inequality, we apply the bound from (He et al., 2023, Theorem 1.1) for the second term, and we use (C.11) for the third term. □

## E  CHOICE OF $d'$

With the error bound displayed in (4.1), we can balance different error terms to choose a $d'$ to attain better accuracy. Meanwhile, it is crucial to ensure that the procedure is still differentially private. Recall $\widehat{\mathbf{M}}$ from Algorithm 2. We can choose that

$$d' := \underset{2 \leq k \leq d}{\arg\min}\left(\sqrt{\sum_{i>k}\sigma_i(\widehat{\mathbf{M}})} + \sqrt{\frac{d}{k}}(\varepsilon n)^{-1/k} + \sqrt{\frac{kd^{2.5}}{\varepsilon n}}\right). \tag{E.1}$$

Firstly, as this definition for $d'$ only depends on the singular values of the private covariance matrix $\widehat{\mathbf{M}}$ and is independent of the true data, we know the output $d'$ is $\varepsilon$-differentially private from the privacy guarantee of $\widehat{\mathbf{M}}$ shown Theorem 2.1.

Moreover, for the accuracy, we have

$$\left|\sum_{i>d'}\sigma_i(\widehat{\mathbf{M}}) - \sum_{i>d'}\sigma_i(\mathbf{M})\right| \leq (d-d')\|\mathbf{A}\|,$$

and hence

$$\sqrt{\sum_{i>d'} \sigma_i(\mathbf{M})} \leq \sqrt{\sum_{i>d'} \sigma_i(\widehat{\mathbf{M}})} + \sqrt{(d-d')\|\mathbf{A}\|} \qquad \text{(E.2)}$$

where $A$ is the Laplacian random matrix defined in Algorithm 2 and with high probability we know $\|A\|$ is bounded by $d^{2.5}/(\varepsilon n)$. Thus such a choice of $d'$ is reasonable.

Finally, after computing the singular values of $\widehat{\mathbf{M}}$, we can compute the choice of $d'$ efficiently within linear time $O(d)$. Therefore (E.1) is a practical method to find a near-optimal hyper-parameter $d'$.

Note that the actual $W_1$ accuracy bound in Theorem 4.2 includes some hidden constants for each term. Also, the inequality (E.2) is far from tight. These are all possible factors may influence the behaviour of our chosen $d'$. Therefore, although such a choice of $d'$ will maintain a low accuracy error bound in principle, we can use it as a reference and find another $d'$ close to it with a relatively large singular value (of $\widehat{\mathbf{M}}$) gap in practice.

## F   OTHER EXPERIMENTS

One of the most essential advantages of synthetic data is the flexibility. Unlike other differentially private algorithms that work for some specific usages, differential private synthetic data allows a wide range of down-streaming statistical tasks while keeping the privacy guarantee. On the dataset of handwritten digits in Section 5, besides the experiment of SVM accuracy, we also includes the errors between the original data and the synthetic data for the mean value and the covariance.

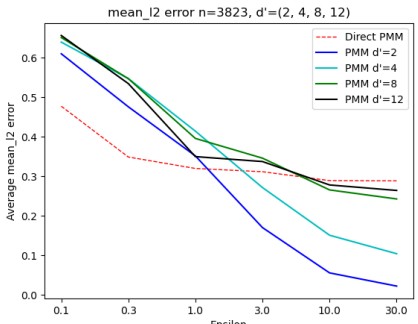
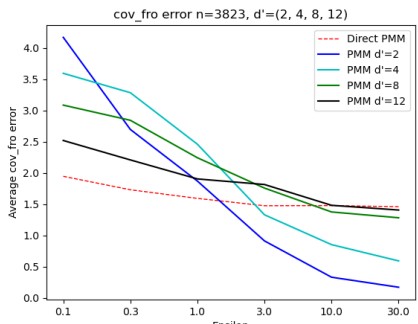

Figure 3: Error of the mean value, in $\ell^2$-norm.

Figure 4: Error of the covariance matrix, in Frobenius norm.

Again, similar to the result in Section 5, the low-dimensional algorithm includes some unnecessary error in the projection step when $\varepsilon n$ is small. When $\varepsilon n$ is large, the projection to a low dimension avoids the curse of high dimensionality and attains better accuracy. Therefore, a potential way to enhance the behavior of the low-dimensional algorithm is to enlarge the dataset. Although applying direct PMM would also benefit from it, but the low-dimensional algorithm achieves a better error rate.

