# OpenReview forum: "Differentially Private Low-dimensional Synthetic Data from High-dimensional Datasets"
_ICLR.cc/2024/Conference — Submitted to ICLR 2024_

### Official Review · Reviewer_SX6K · 2023-10-13

**Soundness:** 3 good
**Presentation:** 3 good
**Contribution:** 2 fair
**Rating:** 3
**Confidence:** 4

**Summary:**

The paper proposes a method to generate DP synthetic data that projects the data into a low-dimensional linear subspace, generates synthetic data there, and projects it back. Projecting to a low-dimensional subspace improves the dimension dependency of the accuracy if the data actually lies close to that subspace. The paper proves a Wasserstein distance accuracy guarantee, and conducts a single experiment on an image dataset.

**Strengths:**

The paper is fairly easy to understand, despite being theory-heavy. Using PCA in this way to get around the curse of dimensionality in DP synthetic data generation is novel to my knowledge.

**Weaknesses:**

## Major Issues
The Laplace mechanism should be introduced in the paper, and its privacy guarantee should be stated. The parameter of the Laplace mechanism should be $\frac{\Delta_1}{\epsilon}$ for $\epsilon$-DP, where $\Delta_1$ is the $L_1$-sensitivity (Dwork and Roth 2014, Definition 3.3). The paper uses the Laplace mechanism to release $\bar{X}$ in Algorithms 1 and 3 with $\Delta_1 = \frac{1}{n}$. However, if the data is $d$-dimensional, the $L_1$ sensitivity of $\bar{X}$ is $\frac{d}{n}$, so it looks like Algorithm 1 does not have the advertised privacy bound. The incorrect sensitivities are used in the accuracy analysis (for example equation C.7), so they are unlikely to be just typos.

The "private covariance matrix" and "private linear projection" steps in Algorithm 1 should have $\epsilon / 4$, not $\epsilon / 2$.

## Minor Issues
The paper should mention that the experiment considers the image labels to be public information in the experiment, so they don't have privacy protection.

The generated images look very poor compared to images showcased in the DP-MERF paper (Harder et al. 2021) for MNIST. As the datasets are different, it would be good to compare with DP-MERF on the dataset used in the paper.

**Questions:**

- You claim that the Wasserstein distance bound gives accuracy guarantees for many machine learning algorithms. How strong are these guarantees in practice? For example, what is the accuracy guarantee for the SVM in the experiment of the paper (if there is one)?
- What is the accuracy of the SVM on the real data, without DP?
- Assuming that the data lies close to a low-dimensional linear subspace (instead of a more general manifold) sounds very restrictive. Do you have some idea on what settings would this assumption be realistic in?
- Is $\sigma_i(M)$ the $i$-th largest eigenvalue in Theorem 1.2, or is the order something else?
- How could $d'$ chosen in practice?

---

> ### Author Response · Authors · 2023-11-20
>
> #### **Weakness**
> Thank you for pointing out the mistake. Indeed, by Laplacian mechanism, to protect $\overline{X}\in \mathbb{R}^d$ with parameter $\varepsilon$ needs entry-wise noises of $\mathrm{Lap}(\frac{d}{\varepsilon n})$, and hence the private mean estimation would have a slightly worse accuracy bound $O(d/n)$. Together with Weakness 2, we will change the privacy budget of each step in Algorithm 1 to have a correct level of privacy, and we will update the result in the simulation experiments. Nevertheless, it would only influence the final result by $O(d/n)$, and this is an inconsequential term compared with the dominating term $n^{-1/d'}$. Therefore, the final accuracy bound Theorem 4.2 for the main low-dimensional synthetic data algorithm remains to be true.
>
> #### **Questions**
> (1) We compare the accuracy of SVM when applied on the true and synthetic data separately. Our experiment suggests the low-dimensional synthetic data approach is useful for classification tasks.
>
> The theoretical utility of Wasserstein bound is guaranteed by the Kantorovich-Rubinstein duality, which ensures the accuracy of all the Lipschitz tasks. (See the last equation in Appendix A.)  As there are alternative variations of SVM, the practical guarantee is out of the scope for this paper.
>
> (2) Without DP requirement, the accuracy of SVM is around 95%.
>
> The experiment in the paper can be seen as a proof of concept for the Wasserstein accuracy bound we proved. As a result, the low-dimensional algorithm does overcome the curse of high-dimensionality.  Optimizing the algorithm more practically could potentially increase SVM's performance. However, it is not the main focus of our work in this paper.
>
> (3) $\sigma_i(M)$ denotes the $i$-th largest singular value of $M$. Here $M$ is semi-positive-definite, so it is also the $i$-th largest eigenvalue.
>
> (4) Thank you for the question. We mentioned the *Adaptive and private choices of $d'$* at Section 1.1, but we omit the details of such optimal choice due to space limitation. We will add the details of the choice of $d'$ in Appendix E. The main idea for the choice of $d'$ is to balance different error terms  Theorem 4.2; we can choose the best $d'$ to minimize the error. Such a minimization problem would be time-efficient as we only need the values of the singular values of the private covariance matrix.

---

> > ### Comment · Reviewer_SX6K · 2023-11-21
> >
> > Thank you for the response. I've increased my score slightly, but I'm still recommending rejection. It is not clear if the linear subspace assumption is useful in any setting, and the results look very poor.

---

> > > ### Author Response · Authors · 2023-11-21
> > >
> > > Thank you for the response. The linear subspace assumptions are common in various foundational theoretical work, such as PCA and linear regression. The main focus of our paper is to prove the theoretical bound of synthetic data accuracy. As a result, with the idea of low-dimensional subspace projection, we deduce a near optimal bound where $n^{-1/d'}$ term is necessary.

---

### Official Review · Reviewer_7H6Q · 2023-10-31

**Soundness:** 3 good
**Presentation:** 3 good
**Contribution:** 2 fair
**Rating:** 6
**Confidence:** 4

**Summary:**

it is known that the expected utility loss of $\varepsilon$-DP synthetic dataset in 1-Wasserstein distance is $\Omega((\varepsilon n)^{-1/d})$. However, if the data lies in a $d'$-dimensional subspace, then one can construct DP synthetic data with $\tilde{O}((\varepsilon n)^{-1/d'})$ utility bound. However, the the previous work's algorithm (Donhauser et al., 2023) has runtime complexity of $O(d^{d'}n + poly(n^{d/d'}))$.

In this work, the authors propose a polynomial-time algorithm for low-dimensional synthetic. The authors also claim that it has a better utility bound that that in Donhauser et al., 2023.

Mainly, the proposed algorithm employs (1) private PCA to project the data to a low-dimensional subspace and (2) hierarchical partition, where Laplacian noise is added to the count in each subregion in order to create a synthetic probability measure..

**Strengths:**

- The authors proposed a polynomial-time algorithm for DP synthetic data that can be more accurate than previous methods if the data is lying in a low dimensional subspace.
- The value of $d'$ can be chosen adaptively by looking at the singular values of the privatized covariance matrix.
- Privacy and utility analysis of the algorithm are provided.

**Weaknesses:**

- The experimental against the full-dimension algorithm is convincing, but I also would like to see how it perform against the method of Donhauser et al. (2023), especially when the authors claim that their method is more accurate than that of the previous work.
- There should be a discussion on the $\sqrt{\sum_{i>d'}\sigma_i(M)}$ term, especially when $d'$ is unknown (which occurs in most use cases). How do we make sure that this term does not dominate the rest of the error bound?

Right now, I am mainly focusing on the soundness of the paper, but I am not fully convinced that the proposed method performs *strictly* better than that of Donhauser et al. (2023). Also see Questions below.

**Questions:**

- In the Conclusion, the authors claim that Donhauser et al. (2023)'s accuracy rate is $(\varepsilon n)^{-1/(d'+1)}$. I skimmed the said paper and couldn't found the exact bound. Can the authors point me to where the said bound is located?
- I think the upper bound can be used to find an "optimal" value of $d'$ by comparing $\sqrt{\sum_{i>d'}\sigma_i(\hat{M})}$ and the rest of term (we also might have to take the bias of $\sigma_i(\hat{M})$ into account. Can the authors make some comments on this approach?

Minor comments:
- In Algorithm 1: "Let $\overline{X}$ be the mean value of the dataset". Is this the original or the synthetic dataset?
- Continuing from above, if it is the mean of the original dataset, can't we just save the privatized mean from the Linear Projection step and add it back after Low-dimensional Synthetic Data? This would help save the privacy budget by $\varepsilon/4$.
- In Algorithm 1, I see two mechanisms with privacy budget $\varepsilon/2$, and two with $\varepsilon/4$. I am not sure if I am interpreting this correctly since total privacy budget is larger than $\varepsilon$.
- In Algorithm 2, the definition of $\boldsymbol{A}_{ij}$ when $i>j$ is missing.

---

> ### Author Response · Authors · 2023-11-20
>
> #### **Weakness**
> (1)  The method of Donhauser et al. (20203) uses the exponential mechanism and it is not very efficient to implement. Our approach is computationally efficient and has a better theoretical guarantee in terms of the error rate. Our main focus is on the theoretical guarantees therefore we do not  implement their methods in this work.
>
> (2) We mentioned the *Adaptive and private choices of $d'$* at Section 1.1, but we omit the details of such optimal choice due to space limitation. The idea is exactly the gist of Question 2: to choose the best $d'$ privately by balancing different terms in the error bound in Theorem 4.2. (And as mentioned in the question, $\sigma_i(\widehat{\mathbf M})$'s are needed to estimate $d'$ privately.) Moreover, such optimization is time efficient. We will add the details of the choice of $d'$ in Appendix E.
>
> #### **Questions**
> (1) The result of $(\varepsilon n)^{-1/(d'+1)}$ is from Theorem 3 in Donhauser et al. (2023). In their settings, the test function is also all the 1-Lipschitz functions, and the dataset is in a low-dimensional manifold with Minkowski dimension $d'$.
>
> (2) Thank you for the suggested approach. We mentioned the *Adaptive and private choices of $d'$* at Section 1.1, but we omit the details of such optimal choice due to space limitation. The idea is exactly the gist of Question 2: to choose the best $d'$ privately by balancing different terms in the error bound in Theorem 4.2. (And as mentioned in the question, $\sigma_i(\widehat{\mathbf M})$'s are needed to estimate $d'$ privately.) Moreover, such optimization is time efficient. We will add the details of the choice of $d'$ in Appendix E.
>
> #### **Minor**
> (1) This is the mean value of the original dataset.
>
> (2) Thank you for the suggestion. Saving the private mean and using it afterward would have better accuracy than adding noises twice. We changed the algorithm to have a more reasonable privacy budget distribution. Nevertheless, it will only influence the final accuracy bound by $O(1/n)$, which is dominated by the $n^{-1/d'}$ term.
>
> (3) Thank you for pointing it out. Combining with the suggestion in Minor Question 2, we reassign the privacy budget in Algorithm 1 in the correct way.
>
> (4) In the algorithm, we define the matrix $A$ to be symmetric, so the entries below the diagonal are implied from the other entries. We will make it clearer in the modification.

---

> ### Comment · Reviewer_7H6Q · 2023-12-03
> **Response**
>
> I thank the authors for addressing all of my questions. I think that the authors have provided sufficient theoretical contribution toward DP synthetic data, but a bit insufficient for practical considerations. I understand that the method from Donhauser et al. (2023) is cumbersome to implement.
>
> In addition to experimenting on downstream tasks, I recommend generating data from a simple model (e.g. well-separated mixture of gaussians) and measure the statistical distance between the model fitted from synthetic data and the original model.
>
> I have raised the score by 1.

---

### Official Review · Reviewer_nzub · 2023-11-03

**Soundness:** 3 good
**Presentation:** 3 good
**Contribution:** 2 fair
**Rating:** 5
**Confidence:** 3

**Summary:**

This paper studies the problem of generating low-dimensional synthetic datasets under the constraint of differential privacy (DP) accurately with respect to the Wasserstein distance. Their algorithms are computationally efficient and make no assumptions on the distribution of the underlying data. They provide both (primarily) theoretical and empirical results to demonstrate their findings.

Their algorithms is decomposed into a few steps. Private PCA followed by projections of data. Then they have an adaptation of the work from He et al (2023). The adding back the private mean vector to shift the subspace correctly.

**Strengths:**

1. The writing of this paper is quite good. The paper was easy enough to follow, and the results were cleanly written.
2. Working in low-dimensions is an important theme for computational and sample efficiency, so their work makes sense in that regime.
3. They don't require the data to have large eigenvalue gaps, so it is general enough by itself.

**Weaknesses:**

1. The expected Wasserstein distance is $poly(d)$. The second term is still alright if we have enough samples, but the third terms is the one that concerns me. When $d$ is large, the third terms does not really help a lot, especially when $d'$ is not too small. That said, the second term becomes too large if $d'$ is large, so there seems to be quite a trade-off over there.
2. More empirical evaluation would have been nice for this venue. I understand that this is mainly a theoretical work, but given that it might have very practical applications, more experimental work would have made sense.
3. The empirical accuracy is not significant when $\varepsilon$ is small, even when as small as $1$, although it does better than a direct application of the work from He et al (2023).

**Questions:**

1. Where are the empirical comparisons with He et al (2023) shown in the paper?
2. At the bottom of page 2, what are $\delta_{X_i}$ and $\delta_{Y_i}$? Should define them somewhere earlier.
3. How does the accuracy improve when the data is actually low-dimensional? It could be approximately low-dimensional, like in Singhal and Steinke (2021) or it could be exactly low-dimensional. How would the results change?
4. How do your results compare with the **lower bounds** on DP synthetic data generation?

---

> ### Author Response · Authors · 2023-11-20
>
> #### **Weakness**
> (1) Yes, there would be a trade-off, and it is possible to choose a better value of $d'$ privately. We mentioned the *Adaptive and private choices of $d'$* at Section 1.1, but we omit the details of such optimal choice due to space limitation. We will add the details of the choice of $d'$ in the appendix. The main idea for the choice of $d'$ is to balance different terms in the error bound of Theorem 4.2, and we can choose the best $d'$ to minimize the error. Such a minimization problem would be time-efficient as we only need the values of the singular values of the private covariance matrix.
>
> For a non-trivial bound of Wasserstein distance, we need the number of data $n$ to be exponentially large w.r.t. $d'$. If $d'$ is too large, one can try to reduce $d'$ as long as the original data has a lower-dimensional structure (in the sense of singular values $\sigma_i(M)$'s). In the case when further decreasing $d'$ is not possible, the lower bound in  (Boedihardjo et al. 2022)  implied that the rate $n^{-1/d'}$ is optimal.
>
> (2) The main focus of our work is on the theoretical side. The experiments can be seen as a proof of concept to overcome the curse of high dimensionality in the Wasserstein accuracy bound. We add more experiments on the basic yet essential statistics of the synthetic data, including mean and covariance.
>
> (3) We agree the empirical accuracy gets worse when $\varepsilon$ is small. We add some explanation for the behavior of low-dimensional algorithm in Section 5. When $\varepsilon$ is small, $(\varepsilon n)^{-1/d'}$ cannot leverage the strength of low-dimensional projection and hence introduce some unnecessary errors. One possible approach is to enlarge the dataset.
>
> Also, optimizing the algorithm more practically could potentially increase its performance. However, it is not the main focus of our work in this paper.
>
> #### **Questions**
> (1) The empirical comparison is in our simulation. The work of He et al. (2023) has no PCA step, and the low-dimensional improvement significantly improves the accuracy and overcomes the curse of high dimensionality.
>
> (2) Thank you for the suggestion. $\delta_{X}$ denote the point mass measure at $X$ and hence $\mu_{\mathcal{X}}$ is the empirical measure for ${\mathcal{X}}$. We will clarify the definition.
>
> (3) Whether the original data is exactly or approximately low-dimensional will only influence the first error term, the singular value term in the main theorem, Theorem 4.2. This term would decrease if the data is near a certain $d'$-dimensional subspace, and it would vanish when the original data is actually low-dimensional. Therefore, one can tell from the singular value term if the current $d'$ is a good choice. We add details in Appendix E about the method to choose $d'$ adaptively and privately.
>
> (4) There isn't a tight lower bound for the private low-dimensional synthetic data. However, the general lower bound of private synthetic data (Boedihardjo et al. 2022) is applicable. When data is actually low-dimensional, our upper bound agrees with the lower bound because one can regard the original data as a dataset on the corresponding subspace. When data is not low-dimensional, by the general lower bound, it is impossible to get a result better than $O(n^{-1/d'})$. However, although the singular value term is reasonable due to the PCA step, it remains unknown whether it is necessary for low-dimensional synthetic data.

---

### Official Review · Reviewer_BUYC · 2023-11-09

**Soundness:** 3 good
**Presentation:** 2 fair
**Contribution:** 2 fair
**Rating:** 5
**Confidence:** 3

**Summary:**

This paper proposes a differentially private algorithm to generate low-dimensional synthetic data with theoretical utility guarantees over the accuracy of the resulting data. Proof-of-concept experiments are given on MNIST, demonstrating the approach can yield synthetic data that provides reasonable utility on downstream tasks.

**Strengths:**

This paper attempts to tackle and important problem which I agree has not been solved by existing DP literature. The theoretical analysis is rigorous and thorough and I have limited criticisms if this is pitched primarily as a theory paper. With that being said, I'm not convinced the most interesting problems in DP synthetic data create are theoretical at this stage.

**Weaknesses:**

My main criticism is that it is unclear how a practitioner looking to use this approach would know whether their data fits the regime in which the utility theorems hold. In particular, the experiments are very limited and in my view provide little information into the practical utility of the approach to most real-world datasets.  For instance it is very unclear that the inferences about the impact of choosing a smaller d' would generalize to other data. In my view the experiments need to be extended to a much wider set of datasets and downstream tasks. Currently the only evaluation is on the accuracy of a prediction, but one of the main benefits of sharing synthetic data is that in principle it could be used for a large stream of data analysis tasks, beyond training ML models (e.g. descriptive stats and regression tasks).

Related to my previous point, why is this the appropriate baseline approach in Figure 2? I would like to see this compared to training the classifier with DP-SGD directly without ever creating the synthetic data for instance and a classifier trained without DP. If the argument is that this approach lends itself to multiple training and analysis tasks, then a broader set of tasks should be included in the experiments.

Finally, I would encourage the authors to include further intuition and descriptions of the algorithms in surrounding text to make the paper more readable. Currently the paper explains what each part of the algorithm does but I feel it lacks explanation of why this is a good idea.

**Questions:**

Why was such a high privacy budget chosen in the experiments? Did you try other values and how did this change the results?

How were hyper-parameters chosen in the experiments?

Have you thought about whether statistical inference is feasible from the output of your algorithm?

---

> ### Author Response · Authors · 2023-11-20
>
> ### **Weakness**
> Thank you for your comments.
>
> (1) The main focus of our work is on the theoretical side.  We do not run extensive experiments to test the utility of the data. The experiment can be seen as a proof of concept for the Wasserstein accuracy bound we proved. As a result, the low-dimensional algorithm does overcome the curse of high-dimensionality and significantly improves the original algorithm in both accuracy and time and space complexity.
>
> (2) Indeed, the goal of our algorithm is to ensure the utility of the synthetic data in various down-streaming tasks. We add the comparison of some more basic yet essential statistics of the dataset, including the mean and the covariance, in Appendix~F.
>
> (3) Thank you for the suggestion. The main idea of our approach is to generate a private subspace and create private coordinates after projecting data onto that private subspace. We add a non-technical summary of the algorithm in the introduction.
>
> ### **Questions**
> (1) We ran the algorithm in different scales of privacy parameter $\varepsilon$ in Figure 2. For Figure 1, we set the parameter to be $\varepsilon=10$ to show that there are still some similarity points between the synthetic data and original data.
>     % When $\varepsilon$ becomes smaller, the synthetic images are no longer recognizable, but training on synthetic data with smaller $\varepsilon$ still has a nontrivial generalization performance, as shown in Figure 2.
>     We change the parameter of Figure~1 with $\varepsilon=4$, which is a reasonable choice in practice, and the accuracy remains good.
>
> (2) Thank you for the question. We mentioned the *Adaptive and private choices of $d'$* at Section 1.1. We will add a more detailed discussion for the choice of $d'$ in the Appendix E. The main idea for the choice of $d'$ is to balance different terms in the error bound of Theorem 4.2, by finding the best trade-off between low-rank approximation error from SVD and error related to privacy guarantee. Such a minimization problem would be time-efficient as we only need the values of the singular values of the private covariance matrix.
>
> (3) Yes, by our Wasserstein distance bound we showed in Theorem 4.2, one can use the synthetic dataset output from Algorithm 1 for mean and covariance estimation and many other statistical inferences by taking a different Lipschitz test function $f$ in the Kantorovich-Rubinstein Duality expression of Wasserstein distance. The information of Kantorovich-Rubinstein duality of $W_1$ is included in Appendix A.

---

> > ### Comment · Reviewer_BUYC · 2023-12-01
> >
> > Thank you for the response. This addresses some of my concerns but I am still unclear of the practical relevance of the results or whether the theoretical results hold in any realistic settings, which I view as the primary shortcoming of this work. As such, I am keeping my score as is.

---

### Meta-Review · Area_Chair_Uck4 · 2023-12-05

**Metareview:**

Summary: the authors propose a differentially private algorithm to generate low-dimensional synthetic data efficiently from a
high-dimensional dataset with a utility guarantee with respect to the Wasserstein distance. The algorithm relies on a private principal component analysis (PCA) procedure with a near-optimal accuracy bound that circumvents the curse of dimensionality.

Strength: Unlike the standard perturbation analysis, their analysis of private PCA works without assuming the spectral gap for the covariance matrix.

Weakness: The experimental results are poor. It is unclear how useful the proposed method is in more practical scenarios.

**Justification For Why Not Higher Score:**

The usefulness of the proposed method seems unclear given the performance of their method tested on a relatively simple scenario.

**Justification For Why Not Lower Score:**

NA

---

### Decision · Program_Chairs · 2024-01-16

Reject